

# Stress drops and earthquake nucleation in the simplest pressure-sensitive ideal elasto-plastic media

Yury Alkhimenkov[1], Lyudmila Khakimova[2,3], and Yury Podladchikov[2,3]

[1]Department of Civil and Environmental Engineering, Massachusetts Institute of Technology, Cambridge, MA 02139, USA
[2]Institute of Earth Sciences, University of Lausanne, Switzerland
[3]Faculty of Mechanics and Mathematics, Lomonosov Moscow State University, Moscow 119991, Russia

**Correspondence:** Yury Alkhimenkov (yalkhime@gmail.com)

**Abstract.** This study explores stress drops and earthquake nucleation within the simplest elasto-plastic media using two-dimensional simulations, emphasizing the critical role of temporal and spatial resolutions in accurately capturing stress evolution and strain fields during seismic cycles. Our analysis reveals that stress drops, triggered by plastic deformation once local stresses reach the yield criteria, reflect fault rupture mechanics, where accumulated strain energy is released suddenly, simu-

lating earthquake behavior. Finer temporal discretization leads to sharper stress drops and lower minimum stress values, while finer spatial grids provide more detailed representations of strain localization and stress redistribution. Our analysis reveals that displacement accumulates gradually during interseismic periods and intensifies during major stress drops, reflecting natural earthquake cycles. Furthermore, the initial wave field patterns during earthquake nucleation are complex, with high-amplitude shear components.

The histogram of stress drop amplitudes shows a non-Gaussian distribution, characterized by a sharp peak followed by a gradual decay, where small stress drops are more frequent, but large stress drops still occur with significant probability. This "solid turbulence" behavior suggests that stress is redistributed across scales, with implications for understanding the variability of seismic event magnitudes.

Our results demonstrate that high-resolution elasto-plastic models can reproduce key features of earthquake nucleation and
stress drop behavior without relying on complex frictional laws or velocity-dependent weakening mechanisms. These findings emphasize the necessity of incorporating plasticity into models of fault slip to better understand the mechanisms governing fault weakening and rupture. Furthermore, our work suggests that extending these models to three-dimensional fault systems and accounting for material heterogeneity and fluid interactions could provide deeper insights into seismic hazard assessment and earthquake mechanics.

## 1 Introduction

Understanding earthquake nucleation remains a significant challenge in geophysics, as it directly influences our ability to predict and mitigate seismic hazards. Earthquake nucleation is often conceptualized through the study of sliding behavior along fault surfaces, with models traditionally describing the interseismic period as one of near-elastic deformation in the surrounding crust, interrupted by phases of anelastic slip that eventually result in seismic rupture (Pranger et al., 2022). Such





models typically rely on phenomenological rate- and state-dependent friction laws (Dieterich, 1978, 1979; Ruina, 1983), which have been highly successful in describing various aspects of the seismic cycle. However, these friction-based models may overlook critical physical processes that govern the transition from aseismic slip to seismic rupture, particularly when plastic deformation and off-fault processes are involved.

Numerical modeling of elasto-plastic behavior has a long history, with early contributions from Cundall (1989, 1990);
Poliakov et al. (1993, 1994); Poliakov and Herrmann (1994). Regularization of strain localization thickness was addressed by Duretz et al. (2019) and de Borst and Duretz (2020). A single-phase (visco)-hypoelastic-perfectly plastic medium was modeled in both 2D and 3D domains by Alkhimenkov et al. (2024c), while compaction-driven fluid flow and shear bands in porous media were numerically modeled in 3D by Alkhimenkov et al. (2024a).

Recent studies have suggested that plasticity plays a crucial role in the nucleation of earthquakes, particularly through off-
fault plasticity mechanisms. Off-fault plasticity refers to the deformation that occurs away from the main fault plane and can significantly influence the dynamics of rupture propagation. Previous works have explored the effects of off-fault plasticity in two-dimensional (2-D) in-plane dynamic rupture simulations (Templeton and Rice, 2008; Kaneko and Fialko, 2011; Gabriel et al., 2013; Tong and Lavier, 2018; Allison and Dunham, 2018). For instance, Dal Zilio et al. (2022) presented a 2-D thermomechanical computational framework for simulating earthquake sequences in a nonlinear visco-elasto-plastic compressible
medium, highlighting the importance of including viscoelastic and plastic behavior in realistic models.

In addition to 2-D studies, three-dimensional (3-D) dynamic rupture simulations incorporating off-fault plasticity have provided deeper insights into the complexity of earthquake mechanics (Wollherr et al., 2018). Another significant advancement was made by Uphoff et al. (2023), who utilized a discontinuous Galerkin method to model earthquake sequences and aseismic slip on multiple faults, demonstrating the versatility of numerical approaches in capturing the nuances of seismic phenomena.

The role of plasticity in earthquake nucleation has also been emphasized in laboratory experiments. Studies have shown that plastic deformation can precede seismic slip, indicating that the onset of plastic yielding may be a precursor to earthquake initiation (Johnson et al., 2008; Scuderi et al., 2016). These experimental findings support the incorporation of plasticity in numerical models to enhance the understanding of the nucleation process.

Furthermore, the impact of material heterogeneity on earthquake dynamics has been investigated extensively. Hetero-
geneities in the crust, such as variations in material properties and fault zone complexity, can influence stress accumulation and release patterns, affecting the timing and magnitude of earthquakes (Ben-Zion and Sammis, 2011; Yao et al., 2017). Numerical studies incorporating these heterogeneities have provided valuable insights into the intricate behavior of fault systems under different loading conditions.

Despite these advancements, there remains a need for simplified models that can effectively capture the essential features
of earthquake nucleation and stress drops while being computationally efficient. The simplest elasto-plastic models offer a promising avenue for such investigations. By focusing on basic physical principles, these models can provide insights into the fundamental mechanisms of earthquake nucleation, such as the role of stress accumulation and release, the interaction between elastic and plastic deformation, and the influence of material heterogeneity on seismic behavior.





In this study, we employ a two-dimensional elasto-plastic model to investigate stress drops and earthquake nucleation. We
conduct a series of numerical simulations to explore the effects of temporal and spatial resolutions on the accuracy of stress
and strain predictions. Our goal is to understand how these resolutions impact the modeled behavior of stress evolution, strain
accumulation, and the nucleation of seismic events. Our approach involves detailed convergence tests for temporal and spatial
discretizations, analysis of stress drop sequences, and examination of interseismic periods. We also investigate the initial wave
field patterns during earthquake nucleation to gain insights into the complex interplay between quasi-static and elasto-dynamic
mechanics. Through this comprehensive study, we aim to highlight the critical role of high-resolution modeling in capturing
the intricate dynamics of earthquake nucleation and stress drops, providing a foundation for future research and practical
applications in seismic hazard assessment.

The novelty of the present study is highlighted by the following contributions:

1. We utilize the simplest pressure-sensitive ideal plasticity model with constant in time and space static friction coefficient.

2. We propose a new physics-based approach explaining spontaneous stress drops in deforming rocks, offering potential applications in modeling earthquake nucleation.

3. We achieve fast computational times using high-resolution models.

## 2 Mathematical formulation

### 2.1 Quasi-statics

The conservation of linear momentum is expressed as:

$$\nabla_j \sigma_{ij} + f_i = 0, \tag{1}$$

where $\sigma_{ij}$ is the stress tensor, $f_i$ is the body force, $\nabla$ is a dell operator, $j = \overline{1..3}$ and Einstein summation convention is applied
(summation over repeated indices). The stress tensor is decomposed into bulk (volumetric) and deviatoric components

$$\sigma_{ij} = -p\delta_{ij} + \tau_{ij}, \tag{2}$$

where $p$ is pressure, $\tau_{ij}$ is the deviatoric stress tensor, $\delta_{ij}$ is the Kronecker delta. The strain rate is defined as

$$\dot{\varepsilon}_{ij} = \frac{1}{2}\left(\nabla_i v_j + \nabla_j v_i\right) \tag{3}$$

The rheology is elasto-plastic, which is characterized by an additive decomposition of the strain rate into an elastic (volumetric
and deviatoric) and plastic components ($\dot{\varepsilon}^{vp} = \dot{\varepsilon}^{vis} + \dot{\varepsilon}^{pl}$)

$$\dot{\varepsilon}_{ij} = \dot{\varepsilon}_{ij}^{eb} + \dot{\varepsilon}_{ij}^{ed} + \dot{\varepsilon}_{ij}^{pl}, \tag{4}$$

where the superscripts $\cdot^{eb}$, $\cdot^{ed}$, $\cdot^{pl}$ denote elastic volumetric (bulk), elastic deviatoric and plastic parts, respectively. The volumetric (bulk) elastic strain rate is

$$\dot{\varepsilon}_{ij}^{eb} = \frac{1}{3}\nabla_k v_k \delta_{ij}, \tag{5}$$



the deviatoric elastic strain rate is

$$\dot{\varepsilon}_{ij}^{ed} = \frac{1}{2G}\frac{\mathcal{D}\tau_{ij}}{\mathcal{D}t}, \tag{6}$$

the deviatoric plastic strain rate is

$$\dot{\varepsilon}_{ij}^{pl} = \dot{\lambda}\frac{\partial Q}{\partial \sigma_{ij}}, \tag{7}$$

where $\dot{\lambda}$ is the plastic multiplier rate and $Q$ is the plastic flow potential. Combining equations (4)-(7), the strain rate can be reformulated as

$$\frac{1}{3}\nabla_k v_k \delta_{ij} + \frac{1}{2G}\frac{\mathcal{D}\tau_{ij}}{\mathcal{D}t} + \dot{\lambda}\frac{\partial Q}{\partial \sigma_{ij}} = \frac{1}{2}\left(\nabla_i v_j + \nabla_j v_i\right) = \dot{\varepsilon}_{ij}. \tag{8}$$

This the system of equation is the static elasto-plastic model routinely used in solid mechanics (Zienkiewicz and Taylor, 2005).

### 2.1.1 Large strain formulation

The inelastic response is described using hypoelastic constitutive theory. Hypoelasticity involves formulating the constitutive equations for stress in terms of objective (frame-invariant) stress rates (de Souza Neto et al., 2011). The rate evolution law for stress is as follows (de Souza Neto et al., 2011; De Borst et al., 2012):

$$\frac{\mathcal{D}\sigma_{ij}}{\mathcal{D}t} = C_{ijkl}^e\dot{\varepsilon}_{kl}^e = C_{ijkl}^e(\dot{\varepsilon}_{kl} - \dot{\varepsilon}_{kl}^{pl}), \tag{9}$$

where $C_{ijkl}^e$ is the elasticity tensor, $\dot{\varepsilon}_{ij} = \dot{\varepsilon}_{ij}^e + \dot{\varepsilon}_{ij}^{pl}$ is the strain rate tensor decomposed into elastic $\dot{\varepsilon}_{ij}^e$ and plastic $\dot{\varepsilon}_{ij}^p$ components. Since our medium is isotropic, the stiffness tensor $C_{ijkl}^e$ can be fully described by the bulk modulus $K$ and shear modulus $G$:

$$C_{ijkl}^e = \left(K - \frac{2}{3}G\right)\delta_{ij}\delta_{kl} + 2G\left(\frac{1}{2}(\delta_{ik}\delta_{jl} + \delta_{il}\delta_{kj})\right). \tag{10}$$

The Jaumann rate of Cauchy stress, represented as $\mathcal{D}\sigma_{ij}/\mathcal{D}t$, is defined by (de Souza Neto et al., 2011):

$$\frac{\mathcal{D}\sigma_{ij}}{\mathcal{D}t} = \frac{\partial \sigma_{ij}}{\partial t} + v_k\frac{\partial \sigma_{ij}}{\partial x_k} - \dot{w}_{ik}\sigma_{jk} - \dot{w}_{jk}\sigma_{ik}, \tag{11}$$

where $\dot{w}_{ij}$ is the vorticity tensor defined as: $\dot{w}_{ij} = \left(\nabla_i v_j - \nabla_j v_i\right)/2$.

## 2.2 Elasto-dynamics

The conservation of linear momentum is given by:

$$\nabla_j\sigma_{ij} + f_i = \rho\frac{\partial v_i}{\partial t}, \tag{12}$$

where $v$ is the velocity and $\rho$ is the density. The stress-strain relation is described by:

$$\frac{\partial \sigma_{ij}}{\partial t} = C_{ijkl}^e\frac{1}{2}\left(\nabla_l v_k + \nabla_k v_l\right), \tag{13}$$

where $i,j,k,l = 1,2,3$.



## 2.3 Plasticity

Plasticity is implemented using a non-associated, pressure-dependent Drucker–Prager criterion (Drucker and Prager, 1952; de Souza Neto et al., 2011; De Borst et al., 2012). According to this criterion, plastic yielding begins when the second invariant of the deviatoric stress, $J_2$, and the pressure (minus the mean stress), $p$, meet the following condition:

$$\sqrt{J_2} - \sin(\varphi)p = \cos(\varphi)c, \tag{14}$$

where $c$ is the cohesion and $\varphi$ is the angle of internal friction. In terms of the stress tensor, plastic deformations occur when the stresses reach the yield surface. The yield function $F$ and the plastic potential $Q$ for the Drucker–Prager criterion are defined as:

$$F(\tau, p) = \sqrt{J_2} - \sin(\varphi)p - \cos(\varphi)c, \tag{15}$$

$$Q(\tau, p) = \sqrt{J_2} - \sin(\psi)p, \tag{16}$$

where $\psi \leq \varphi$ is the dilation angle. In two dimensions under plane strain conditions, with $\sigma_{zz} = \frac{1}{2}(\sigma_{xx} + \sigma_{yy})$, the Drucker–Prager criterion is equivalent to the Mohr-Coulomb criterion (Templeton and Rice, 2008). In 2-D, the second invariant of the deviatoric stress, $J_2$, is expressed as:

$$J_2 = \frac{1}{2}\tau_{ij}\tau_{ji} = \frac{1}{2}(\tau_{xx}^2 + \tau_{yy}^2) + \tau_{xy}^2. \tag{17}$$

As long as $F \leq 0$, the material remains in the elastic regime. Once $F$ reaches zero ($F = 0$), plasticity is activated. If the material remains in a plastic state ($\partial F/\partial t = 0$), plastic yielding continues. The current implementation of perfect plasticity requires small time increments and is computationally expensive. To ensure spontaneous strain localization, strain softening is often introduced, which promotes the formation of shear bands (Lavier et al., 1999; Moresi et al., 2007; Popov and Sobolev, 2008; Lemiale et al., 2008). However, there are concerns about the thermodynamic admissibility of such solutions (Duretz et al., 2019). Additionally, the softening or hardening moduli are small compared to the shear modulus and can be neglected as a first-order approximation, leading to the ideal plasticity model used in the present study.

## 3 Numerical implementation

### 3.1 Discretization

The numerical domain $V$ is discretized using a staggered grid in both space and time (Virieux, 1986). This method provides a variant of the conservative finite volume approach (Dormy and Tarantola, 1995). The total number of grid cells is limited only by the available GPU memory. For the elasto-dynamic equations, an explicit time integration method is employed, offering second-order accuracy in both space and time. Detailed representations of the discrete equations are available in Alkhimenkov





et al. (2021). For the quasi-static equations, the discrete scheme achieves second-order accuracy in space. Advection is carried
out using an upwind scheme, which is first-order. Consequently, due to the application of the pseudo-transient method, the
solution demonstrates first-order accuracy in time (Alkhimenkov and Podladchikov, 2024).

### 3.2 Accelerated pseudo transient method

The solution of the quasi-static equations is achieved using the matrix-free accelerated pseudo-transient (APT) method (Frankel,
1950; Räss et al., 2022; Alkhimenkov and Podladchikov, 2024). The core concept of this method involves solving dynamic
equations with appropriate attenuation of the dynamic fields instead of directly solving inertialess equations. Once the dynamic
fields attenuate to a specific precision (e.g., to $10^{-12}$), the solution of the quasi-static equations is attained. In other words, the
quasi-static problem serves as an attractor for the dynamic problem with damping. The APT method is capable of handling
numerical domains with more than a billion grid cells. Additionally, since all operations are local, this method can be naturally
parallelized using GPUs, which is the approach taken in this study.

### 155 3.3 Implementation of plasticity

In the return mapping algorithm, the following steps are performed:

1. Calculate the components of the trial deviatoric stresses, $\tau_{ij}^{\mathrm{trial}}$. 2. Compute the trial second invariant of the deviatoric
stresses, $J_2^{\mathrm{trial}}$, using $\tau_{ij}^{\mathrm{trial}}$. 3. Determine $F^{\mathrm{trial}}$ using the equation:

$$F^{\mathrm{trial}} = \sqrt{J_2^{\mathrm{trial}}} - (\sin(\varphi)p + \cos(\varphi)c). \tag{18}$$

When the material is in the plastic state, the trial deviatoric stress components, $\tau_{ij}^{\mathrm{trial}}$, are re-scaled according to:

$$\tau_{ij}^{\mathrm{new}} = \tau_{ij}^{\mathrm{trial}} \left( 1 - \frac{F^{\mathrm{trial}}}{\sqrt{J_2^{\mathrm{trial}}}} \right) \equiv \tau_{ij}^{\mathrm{trial}} \tilde{\lambda}, \tag{19}$$

where $\tilde{\lambda} = 1 - \frac{F^{\mathrm{trial}}}{\sqrt{J_2^{\mathrm{trial}}}}$ is the scaling parameter. This re-scaling process is iterated over "pseudo-time" until the updated trial de-
viatoric stresses, $\tau_{ij}^{\mathrm{new}}$, satisfy the plasticity criterion, ensuring $F^{\mathrm{trial}} = 0$ (thus, $\tilde{\lambda} = 1$ and no re-scaling occurs). A regularized
version of this procedure modifies formula (19) to (assuming non-zero dilatation angle $\psi$):

$$\tilde{\lambda} = 1 - \frac{F^{\mathrm{trial}} \Delta t G^{\mathrm{e}}}{\sqrt{J_2}(G^{\mathrm{e}} \Delta t + K \Delta t \sin\varphi \sin\psi + \eta^{\mathrm{vp}})}. \tag{20}$$

where $\eta^{\mathrm{vp}}$ is the viscosity of the damper (regularization parameter). This implementation of plasticity through re-scaling
deviatoric stress components is equivalent to the standard procedure using the plastic multiplier rate, $\dot{\lambda}$, which is defined as

$$\dot{\lambda} = \frac{F^{\mathrm{trial}}}{\Delta t G^{\mathrm{e}} + K \Delta t \sin\varphi \sin\psi + \eta^{\mathrm{vp}}}. \tag{21}$$

For a more detailed explanation of how plasticity with regularization is implemented in single-phase media, refer to Duretz
et al. (2018, 2019, 2021).



### 3.4 Nondimensionalization

We select the following dimensionally independent scales: length $l^* = L_x$, time $t^* = 1/a$, and pressure $p^* = G_0$. Here, $L_x$ represents the size of the computational domain in the $x$-dimension, and $a$ denotes the background strain rate. Deformation occurs over times inversely proportional to the initial background strain rate $a_0$ at $t = 0$. The ratio of cohesion $c$ to the pressure scale $p^*$ is defined as $r = \frac{c_0}{G_0}$.

### 3.5 Model Configuration and Boundary conditions

The computational domain is a square with dimensions $x, y \in [0, L_x] \times [0, L_y]$. All simulations presented in this study have been performed using a simple initial model configuration. The coefficient of internal friction $\mu = 0.6$ in all computations. The pure shear boundary conditions are applied by prescribing velocities at all boundaries

$$v_x = ax \tag{22}$$

and

$$v_y = -ay, \tag{23}$$

which corresponds to the extension in x-dimension and compression in y- dimension. We impose loading increments applied to the strain components. At all boundaries, free-slip boundary conditions are implemented. The following initial conditions are implemented:

$$p = 0.004, \tag{24}$$

$$\tau_{xx} = 0.012, \tag{25}$$

$$\tau_{yy} = -0.012. \tag{26}$$

We set anomalies to the non-dimensional cohesion $c$ which has an upside-down Gaussian distribution with the lowest value in the center of the model. The expression is the following:

$$c = 0.012 + 0.0005 \, exp(-(x/0.2)^2 - (y/0.2)^2). \tag{27}$$

## 4 Results

### 4.1 Low resolution simulation

In our low-resolution simulations (Figure 2), we observe that the evolution of the integrated stress $\sigma_{xx}^{\text{INT}}$ follows a monotonic trend, with no discernible stress drops. This outcome aligns with the coarse spatial discretization of $N = 63^2$ grid cells, wherein



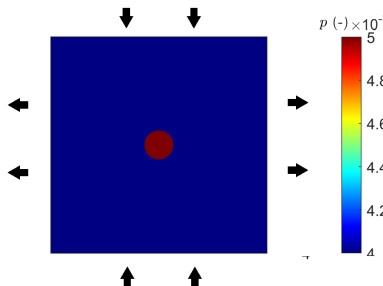

**Figure 1.** Heterogeneous initial setup of cohesion $c$. The arrows indicate the pure shear boundary condition which is applied at the model boundaries.

finer stress and strain variations cannot be accurately captured. The strain localization patterns observed in the pressure and strain rate fields remain symmetric throughout the loading process, indicating a relatively uniform distribution of deformation.

This result is expected in low-resolution simulations, where the model's ability to resolve localized strain structures, such as shear bands or deformation zones, is limited by the grid resolution.

Moreover, in low-resolution models, the regularization parameter $\eta^{\mathrm{vp}} = 6 \times 10^{-4}$ plays a critical role in smoothing out any potential irregularities in the stress field. While this helps stabilize the model numerically, it also suppresses any potential stress drops. The absence of stress drops in low-resolution simulations suggests that grid refinement is necessary to capture

more detailed stress and strain distributions, which could better reflect the underlying physical processes driving seismicity.

## 4.2 Sufficient resolution simulation

In contrast, our sufficient resolution simulations with $N = 1023^2$ grid cells reveal several significant stress drops (Figure 3), suggesting that the model resolution is now capable of accurately capturing the dynamic changes in stress during loading. The regularization parameter is set to $\eta^{\mathrm{vp}} = 1 \times 10^{-5}$. The stress drops correspond to instances of rapid strain localization,

where non-symmetric shear bands develop and propagate throughout the model domain. These shear bands form due to the onset of plastic yielding, which is triggered as local stresses surpass the material's yield strength. This non-symmetric strain localization is a hallmark of plastic deformation and closely resembles the behavior observed in laboratory experiments on rock deformation, where similar patterns of localized shear zones have been reported (Johnson et al., 2008; Scuderi et al., 2016).

Furthermore, the higher resolution provides clearer insights into the spatial structure of the stress and strain fields, revealing

the complex, non-uniform distribution of deformation during earthquake nucleation. Notably, the stress drops become more pronounced and sharper as the temporal resolution is increased, underscoring the importance of both spatial and temporal refinement in accurately capturing the dynamics of stress accumulation and release.



**Figure 2.** Low resolution simulation. Panel (a) shows the integrated stress $\sigma_{xx}^{\mathrm{INT}}$ versus strain increments, panels (b-d) show pressure $p$ for at three different stages of the simulation, panels (e-g) show $\log_{10}\epsilon_{II}$ and panels (h-j) show $\log_{10}\dot{\epsilon}_{II}$.





### 4.2.1 Mohr's circle analysis

Figure 4 illustrates the stress state of the material through a Mohr's circle representation at two critical stages: the beginning of
loading (panel (a)) and after shear band localization (panel (b)). Mohr's circle is a graphical tool used to represent the state of
stress at a point, plotting the normal and shear stresses acting on different planes through that point.

In panel (a), the stress is evenly distributed, and the circle lies within the elastic regime, meaning the material is undergoing
purely elastic deformation. The material is in equilibrium, with no plastic yielding or localized strain. This initial Mohr's
circle is small, reflecting the lower stress magnitudes early in the loading process. The friction coefficient is $\mu = 0.6$, which
corresponds to $\bar{\varphi} \approx 31°$. Note that $\tan(\bar{\varphi}) = \tan(31°) \approx 0.6$.

Panel (b) shows Mohr's circle after the onset of strain localization, coinciding with a significant stress drop in the simulation.
As loading progresses and the material begins to yield, Mohr's circle shifts, reflecting the decrease in pressure as the system
approaches the yield criterion. The expansion of Mohr's circle towards the yield envelope indicates that the material has reached
its plastic limit, and shear bands begin to localize. We also highlight additional intersections along the yield envelope.

Of particular importance is the angle $\varphi_A \approx 27°$ (estimated numerically by analyzing the green triangle, Figure 4b), which is
the apparent angle related to the *apparent coefficient of friction* $\mu_a$. According to Byerlee and Savage (1992), the value of $\mu_a$
is always lower than the real coefficient of internal friction, $\mu$. There is a theoretical formula that relates $\mu_a$ and $\mu$ (Byerlee and
Savage, 1992):

$$\mu_a = \sin(\tan^{-1}\mu) \equiv \sin(\tan^{-1}0.6) \approx 0.51 \tag{28}$$

The theoretical value of $\mu_a = 0.51$ provides us with $\varphi_A = \tan^{-1}(0.51) \approx 27°$, which is the same value estimated numeri-
cally by analyzing the green triangle (see paragraph above and Figure 4b). The reduced apparent coefficient of friction is a
consequence of plastic flow in the fault gouge, which allows slip to occur more easily along the fault plane, despite the actual
slip occurring along the Coulomb shear planes. Overall, panel (b) emphasizes how localized plastic shear flow within the fault
gouge governs fault slip, offering insights into the mechanics of fault weakening.

### 240 4.3 Focusing on stress drops and interseismic period

Figure 5 shows the numerical simulation with focus on stress drops and interseismic periods. For example, $u_x$ corresponding
to stress drop 1 is calculated as $u_x = u_x(t_2) - u_x(t_1)$, where $t_1$ corresponds to the total strain just before the stress drop (red
circle) and $t_1$ corresponds to the total strain just after the stress drop (second red circle). The calculation of $u_x$ corresponding
to the interseismic period 1 is similar: $u_x = u_x(t_3) - u_x(t_2)$, where $t_2$ corresponds to the total strain just after the stress drop
1 (second red circle) and $t_3$ corresponds to the total strain just before the stress drop 2 (third red circle).







**Figure 3.** Sufficient resolution simulation. Panel (a) shows the integrated stress $\sigma_{xx}^{\text{INT}}$ versus strain increments, panels (b-d) show pressure $p$ for at three different stages of the simulation, panels (e-g) show $\log_{10}\epsilon_{II}$ and panels (h-j) show $\log_{10}\dot{\epsilon}_{II}$.

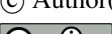





**Figure 4.** Mohr's circle. Panel (a) shows the initial Mohr's circle at the beginning of loading. Panel (b) shows the Mohr's circle after shear band localization with additional intersections.



**Figure 5.** Stress drops and interseismic periods: numerical simulation of compressible elasto-plastic equations with the resolution of $N = 1023^2$ grid cells. Panel (a) shows the integrated stress $\sigma_{xx}$ versus strain increments. Panels (b,d,f,h) show displacement increments $\Delta u_x$ corresponding to stress drops and interseismic periods. Panels (c,e,g,i) show pressure increments $\Delta u_x$ corresponding to stress drops and interseismic periods.



## 4.4 Temporal convergence tests

First, we conduct a temporal convergence test (Figure 6). Using a spatial resolution of $N = 511^2$ grid cells, simulations are performed with strain increments $\Delta\epsilon_{xx} = 1 \times 10^{-5}$, $\Delta\epsilon_{xx} = 4 \times 10^{-5}$, $\Delta\epsilon_{xx} = 10 \times 10^{-5}$. It is observed that the evolution of the integrated stress $\sigma_{xx}^{\text{INT}}$ with strain increments is converging to a specific pattern as the number of increment increasing.
Simulations with finer temporal discretization result in slightly sharper drops in $\sigma_{xx}^{\text{INT}}$.

## 4.5 Spatial convergence tests

To investigate the dependence of the integrated stress $\sigma_{xx}$ versus spatial resolution, we conduct experiment with spatial discretizations of $N = 63^2$, $N = 1023^2$, and $N = 2047^2$ (Figure 7). It can be seen that the low resolution simulation $N = 63^2$ does not produce any stress drops. However, simulations with sufficient resolution produce stress drops and their amplitudes
are similar.

## 4.6 Effect of the regularization

To illustrate the dependence of stress drop amplitude versus regularization, we conduct one more series of computations (Figure 8). The spatial discretizations is the same in all simulations $N = 511^2$ but the regularization viscosity $\eta^{\text{vp}}$ is different. Note the lower regularization leads to the more pronounced stress drops as can be seen in Figure 8. Too high regularization may
completely miss the stress drop (Figure 8).

Figure 9 shows the simulation results for different spatial discretizations of $N = 63^2$, $N = 255^2$, and $N = 1023^2$ grid cells. Due to high regularization, the results are identical and the thickness of the shear bands is the same in all panels. However, due to over-regularization, the stress drop is not visible.







**Figure 6.** Temporal convergence tests. Panel (a) shows the integrated stress $\sigma_{xx}^{\text{INT}}$ versus strain increments. Panels (b,c,d,e) show displacement increments $\Delta u_x$ corresponding to stress drops and interseismic periods that correspond to $\Delta \epsilon_{xx} = 1 \times 10^{-5}$. Panels (f,g,h,i) show pressure increments $\Delta u_x$ corresponding to stress drops and interseismic periods that correspond to $\Delta \epsilon_{xx} = 4 \times 10^{-5}$. Panels (j,k,l,m) show pressure increments $\Delta u_x$ corresponding to stress drops and interseismic periods that correspond to $\Delta \epsilon_{xx} = 10 \times 10^{-5}$







**Figure 7.** Spatial convergence tests. Panel (a) shows the integrated stress $\sigma_{xx}^{\text{INT}}$ versus strain increments. Panels (b,c,d,e) show displacement increments $\Delta u_x$ corresponding to stress drops and interseismic periods that correspond to $N = 63^2$ grid cells. Panels (f,g,h,i) show pressure increments $\Delta u_x$ corresponding to stress drops and interseismic periods that correspond to $N = 1023^2$ grid cells. Panels (j,k,l,m) show pressure increments $\Delta u_x$ corresponding to stress drops and interseismic periods that correspond to $N = 2047^2$ grid cells.



**Figure 8.** Effect of the regularization. Panel (a) shows the integrated stress $\sigma_{xx}^{\mathrm{INT}}$ versus strain increments. Panels (b,c,d,e) show displacement increments $\Delta u_x$ corresponding to stress drops and interseismic periods that correspond to $\eta^{\mathrm{vp}} = 0$. Panels (f,g,h,i) show pressure increments $\Delta u_x$ corresponding to stress drops and interseismic periods that correspond to $\eta^{\mathrm{vp}} = 1 \times 10^{-5}$. Panels (j,k,l,m) show pressure increments $\Delta u_x$ corresponding to stress drops and interseismic periods that correspond to $\eta^{\mathrm{vp}} = 10 \times 10^{-5}$. The resolution is $N = 511^2$ grid cells in all simulations).



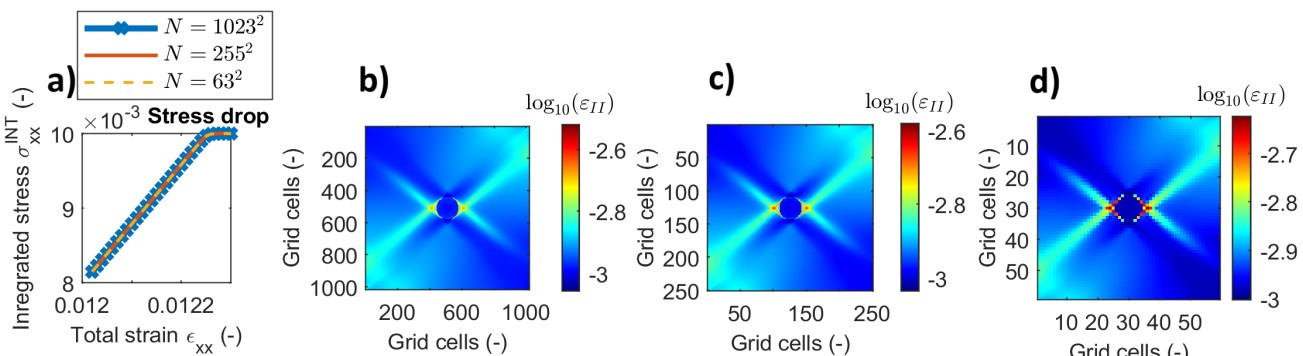

**Figure 9.** Effect of the regularization: spatial convergence loading increments. Panel (a) shows the integrated stress $\sigma_{xx}^{\text{INT}}$ versus time, panels (b-d) show $\log_{10} \epsilon_{II}$ for a set of different spatial discretizations: $N = 63^2$, $N = 255^2$, and $N = 1023^2$ grid cells.



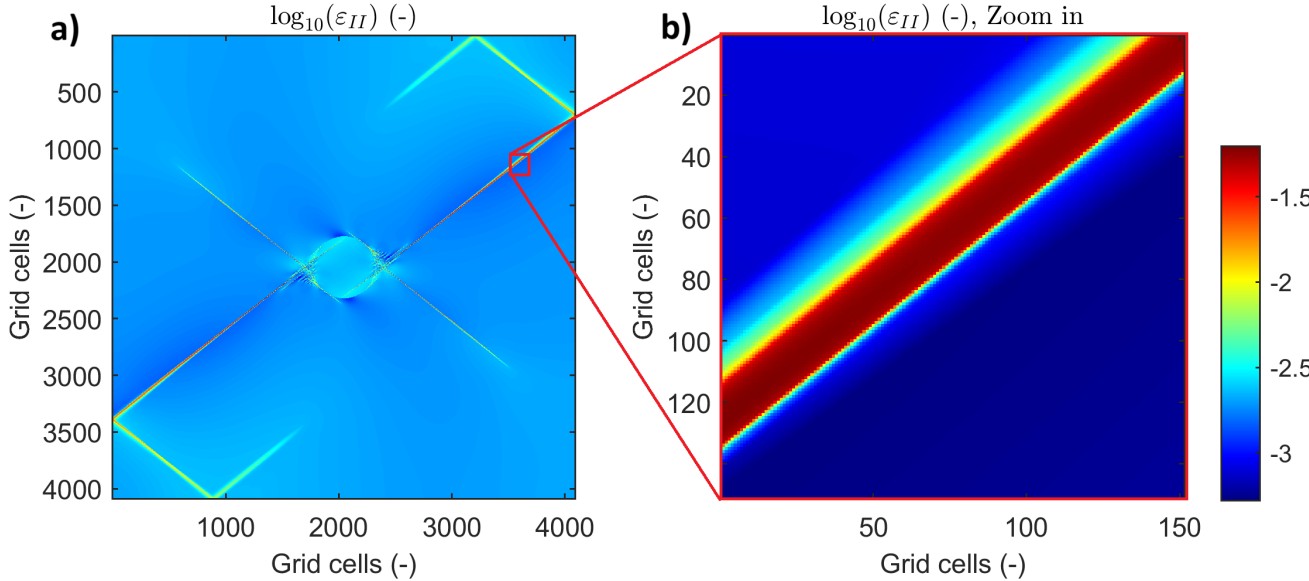

**Figure 10.** Panel (a) shows $\log_{10} \epsilon_{II}$. Panels (b) corresponds to zoom-in focusing on a single shear band thickness.

### 4.7 High-resolution simulation

The high-resolution simulation is performed with a grid size of $N = 4095^2$, allowing for the capture of finer details in the stress, strain, and pressure fields. In these simulations, the component $\log_{10} \epsilon_{II}$ is used to represent the second invariant of the deviatoric strain rate tensor, which highlights zones of intense strain localization, typically corresponding to regions where shear bands form.

As shown in Figure 10, the regularization applied is sufficient to resolve pressure drops and observe localized strain structures 270 across multiple grid cells. The use of such a fine grid provides enhanced spatial resolution, allowing us to capture more realistic patterns of strain localization that resemble those seen in natural seismic zones, where shear bands and strain concentrations often precede fault rupture or failure events.

Additionally, Figure 11 presents a zoomed-in view of these strain localization features as a 3D plot. In this plot, the vertical axis (z-dimension) corresponds to the amplitude of the pressure field $p$ (Figure 11a) and the $\log_{10} \epsilon_{II}$ (Figure 11b). The 275 localized shear band thickness is clearly visible, indicating zones of intense deformation and stress concentration. Such fine-scale detail is critical for accurately modeling the mechanics of earthquake nucleation, where small variations in stress and strain fields can have significant impacts on rupture initiation and propagation.

The high-resolution results emphasize the importance of spatial resolution in capturing the complex interplay between elastic and plastic deformation, stress accumulation, and release mechanisms, which are crucial for understanding the nucleation of 280 seismic events.



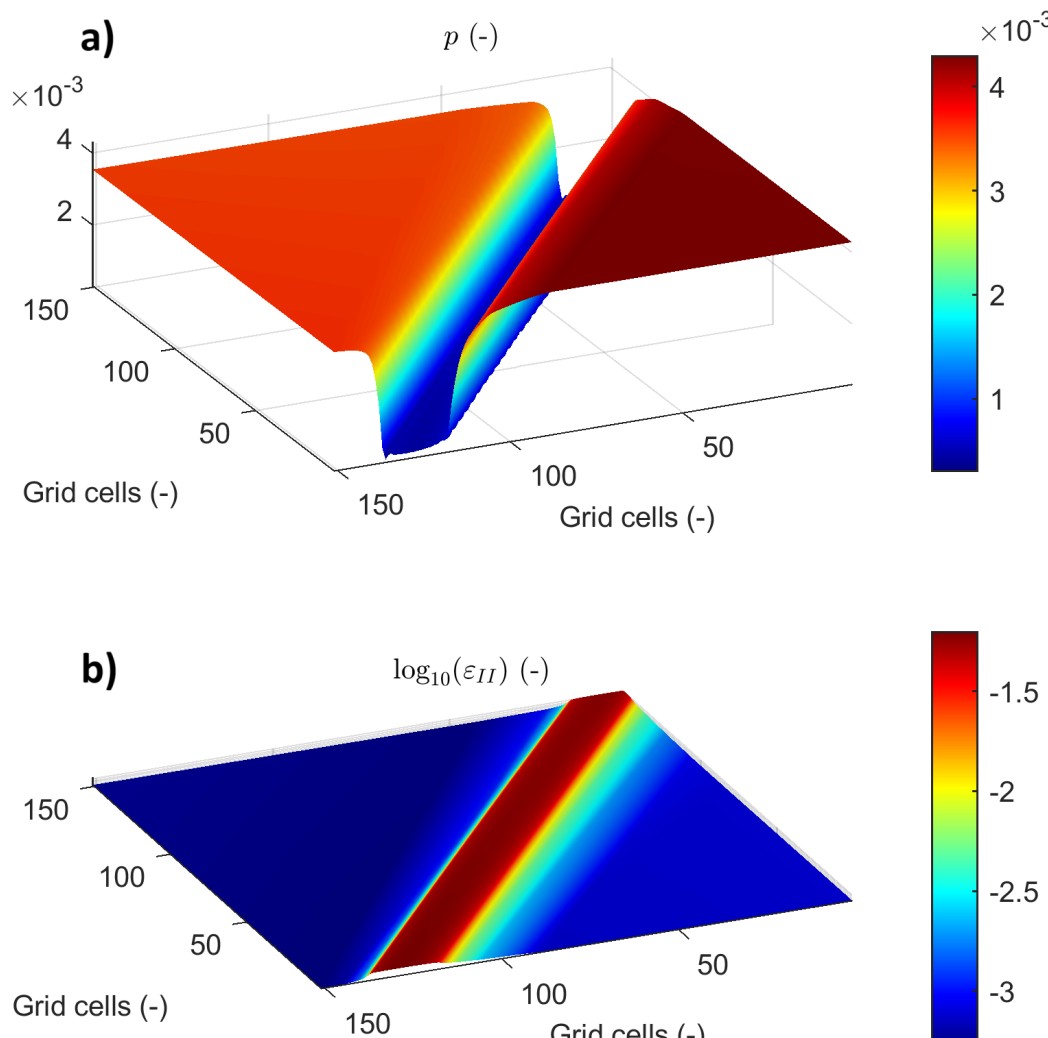

**Figure 11.** Zoom-in result for the spatial discretizations of $N = 4095^2$ as a 3D plot where z-dimension corresponds to the amplitude of the pressure field, $p$ (panel a) and $\log_{10} \epsilon_{II}$ (panel b).





## 4.8 Dilatancy

Dilatancy refers to the volumetric expansion that occurs in a material when it undergoes shear deformation, particularly under conditions of plastic flow. This phenomenon is especially important in the study of earthquake mechanics, as it affects the porosity and fluid flow within fault zones, and hence, influences the strength and failure behavior of the fault.

In Figure 12, numerical simulations are performed for different values of the dilatation angle $\psi$, representing different degrees of dilatancy in the material. The results illustrate the effect of the dilatation angle on strain localization and fault weakening. For a small dilatation angle ($\psi = 5°$), the material exhibits relatively limited volumetric expansion during shear deformation, leading to more localized strain and narrower shear bands. This corresponds to less energy dissipation and a more brittle-like failure response.

As the dilatation angle increases to a moderate value ($\psi = 15°$), the material shows more volumetric expansion, which slightly widens the shear bands and leads to a more diffuse strain localization pattern. This represents a more ductile response, where plastic deformation is distributed over a broader zone.

In the case of $\psi = 30°$, corresponding to the associated plasticity model, the volumetric expansion is maximized, and the shear bands become much broader. This behavior reflects greater energy dissipation, as the material undergoes significant

volumetric changes during shear deformation. The associated plasticity model is typically used for materials that exhibit a strong dilatant response, such as certain types of granular soils or fractured rocks.

These results highlight the sensitivity of fault behavior to the dilatation angle, emphasizing the importance of including dilatancy effects in models of fault mechanics and earthquake nucleation. In real fault zones, dilatancy can influence pore fluid pressure and, consequently, the effective normal stress, which plays a crucial role in controlling fault strength and stability.

For comparison, Figure 13 shows numerical results for $\psi = 0$ and $\varphi = 0$, which correspond to the plasticity behavior typically observed in metals. In this case, the material does not exhibit any volumetric expansion during shear deformation, leading to purely deviatoric plastic flow. The absence of dilatancy results in narrow and highly localized shear bands, as expected in materials that do not undergo volumetric changes. This behavior is characteristic of metals under plastic deformation, where energy dissipation is minimized, and the material response remains predominantly brittle. These results underscore the con-

trasting effects of dilatancy on shear band formation and highlight the unique deformation mechanisms in metallic versus dilatant materials.

## 4.9 Stress drop sequence

Figure 14 presents the integrated stress $\sigma_{xx}^{\mathrm{INT}}$ versus time (Figure 14a) for different temporal resolutions. Figure 14b shows the zoomed in plot where sharp stress drops can be visible. The stress drops exhibit varying magnitudes and irregular spacing. The

simulation with fine temporal resolution and the lowest regularization corresponds to the sharpest stress drops (blue curve). The simulations with low temporal resolution does not present a proper stress drops.

During the loading process, numerous stress drops are observed (Figure 14). These stress drops correspond to sudden shifts in the system's stress state, where the strain localization reaches a critical point, and further deformation in the prescribed



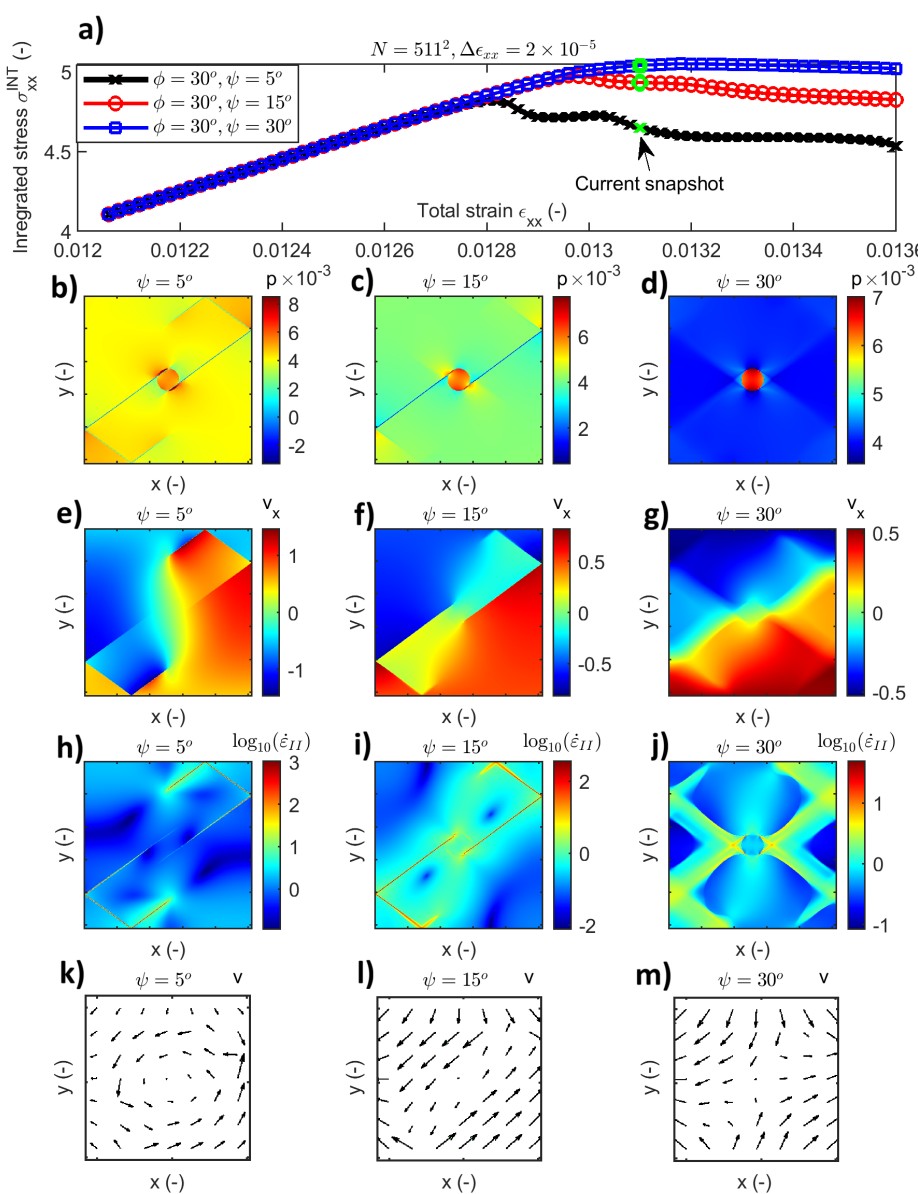

**Figure 12.** Numerical results for different angles $\psi$, where $\psi = 5$ corresponds to the small dilatation angle, $\psi = 5$ corresponds to the moderate dilatation angle, and $\psi = 30$ corresponds to the associated plasticity.



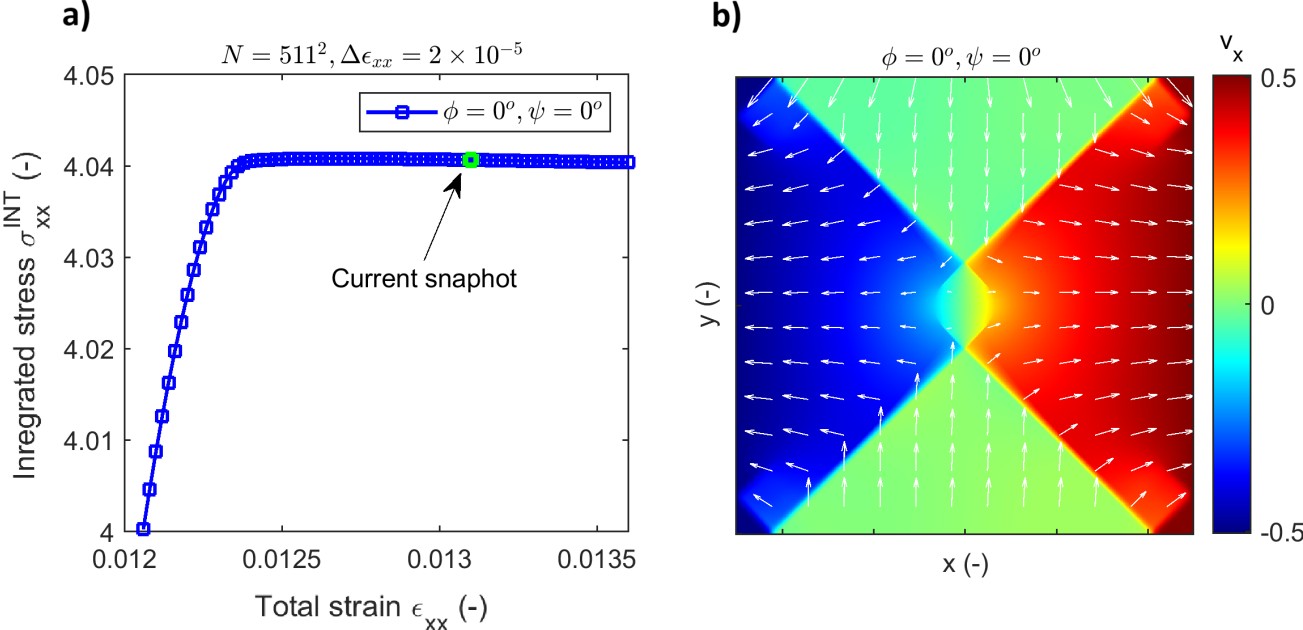

**Figure 13.** Numerical results for $\psi = 0$ and $\varphi = 0$, corresponding to the plasticity behavior of metals.

direction becomes untenable. As a result, the system undergoes a rapid redistribution of stress, manifested as a drop in the

integrated stress $\sigma_{xx}^{\text{INT}}$.

These stress drops are indicative of dynamic rupture events, akin to the rapid stress release observed during seismic slip in natural earthquakes. The sequence of stress drops observed in the simulation resembles the cyclic behavior of fault systems, where interseismic periods of stress accumulation are interrupted by seismic events. By increasing the temporal resolution, we capture sharper, more distinct stress drops, highlighting the need for high-resolution models to accurately represent seismic

processes.

### 4.9.1 Interseismic period and stress drops

Figure 15 illustrates the displacement increments $\Delta u_x$ during the interseismic period (Figures 15a-b) and the stress drops (Figures 15c-d). For instance, Figure 15a shows the displacement increment $\Delta u_x = u_x(3) - u_x(2)$, where $u_x(2)$ and $u_x(3)$ represent the displacement fields at the beginning and end of the interseismic period, respectively (the period between two high-

amplitude stress drops. Similarly, the displacement increments $\Delta u_x = u_x(2) - u_x(1)$ during major stress drops are shown in Figures 15c-d. It is evident that displacement accumulates during the interseismic period (without major stress drops) and also intensifies during major stress drops.

Our simulation results also demonstrate the behavior of the material during interseismic periods, where displacement gradually accumulates without significant stress drops. As shown in Figure 15, displacement increments $\Delta u_x$ during interseismic





**Figure 14.** Numerical simulation of elasto-plastic equations with the resolution of $1023^2$ grid cells for 2500 loading increments in time. Panel (a) shows the integrated stress $\sigma_{xx}^{\mathrm{INT}}$ versus loading increments. Panel (b) shows the integrated stress $\sigma_{xx}^{\mathrm{INT}}$ versus loading increments for a portion of the full model. Panel (b) shows the final strain localization pattern for all three simulations.





periods increase progressively as loading continues, leading up to the next stress drop event. This behavior mirrors the slow, aseismic slip observed between seismic events in fault zones. The gradual buildup of displacement during interseismic periods corresponds to the elastic loading of the fault system, while the rapid displacement during stress drops corresponds to seismic slip.

These findings suggest that the interaction between elastic and plastic deformation plays a critical role in controlling the 335 timing and magnitude of seismic events. The gradual accumulation of displacement during the interseismic period reflects the fault's capacity to store elastic strain energy, which is then rapidly released during seismic events, leading to a stress drop.

### 4.9.2 Histogram of stress drop amplitudes

The histogram of stress drop amplitudes shown in Figure 16 provides a quantitative representation of the frequency and magnitude of stress drops occurring during the simulations. The distribution of stress drop amplitudes is notably non-Gaussian, 340 characterized by a sharp peak followed by a gradual decay, indicating that while small stress drops are more common, larger stress drops still occur with significant probability. This distribution resembles what is often observed in turbulent systems, where a few large events (bursts) coexist with numerous smaller fluctuations, a phenomenon referred to as "solid turbulence."

Figure 16b presents three histograms of stress drop amplitudes corresponding to very small strain increments, providing high-resolution data. The overall pattern across the histograms is similar, indicating the convergence of our results. In Fig-345 ure 16c, two histograms are shown: one for a high-resolution loading increment ($\Delta\epsilon_{xx} = 1 \times 10^{-5}$) and the other for the lowest resolution used in this study ($\Delta\epsilon_{xx} = 100 \times 10^{-5}$). The difference between these two is substantial, demonstrating that the low-resolution case fails to capture the full spectrum of stress drop amplitudes. Lastly, Figure 16d compares four histograms at high, intermediate, and low resolutions for further comparison.

In the context of our elasto-plastic model, the non-Gaussian nature of the histogram suggests a complex interaction between 350 localized plastic yielding and the broader elastic response of the material. Just as in fluid turbulence, where energy cascades from large to small scales, in solid turbulence, stress is redistributed across different spatial and temporal scales, leading to a range of stress drop magnitudes. This complex behavior highlights the inherent intermittency and unpredictability in the system's response, where stress accumulates gradually but is released in sudden, sporadic bursts during stress drops.

Understanding the solid turbulence-like behavior in these systems is crucial for developing accurate models of seismicity, 355 where stress drops correspond to earthquake events. The implications of this behavior are significant for earthquake hazard assessment, as it suggests that a wide range of earthquake magnitudes should be expected, with smaller events being far more frequent than larger ones. This insight aligns with the Gutenberg-Richter law, which describes the frequency-magnitude distribution of earthquakes but from a plastic deformation perspective. Overall, the histogram of stress drop amplitudes reinforces the idea that the simplest elasto-plastic models, despite their minimalistic assumptions, are capable of capturing complex, 360 emergent behavior that is often associated with more sophisticated models of seismicity.





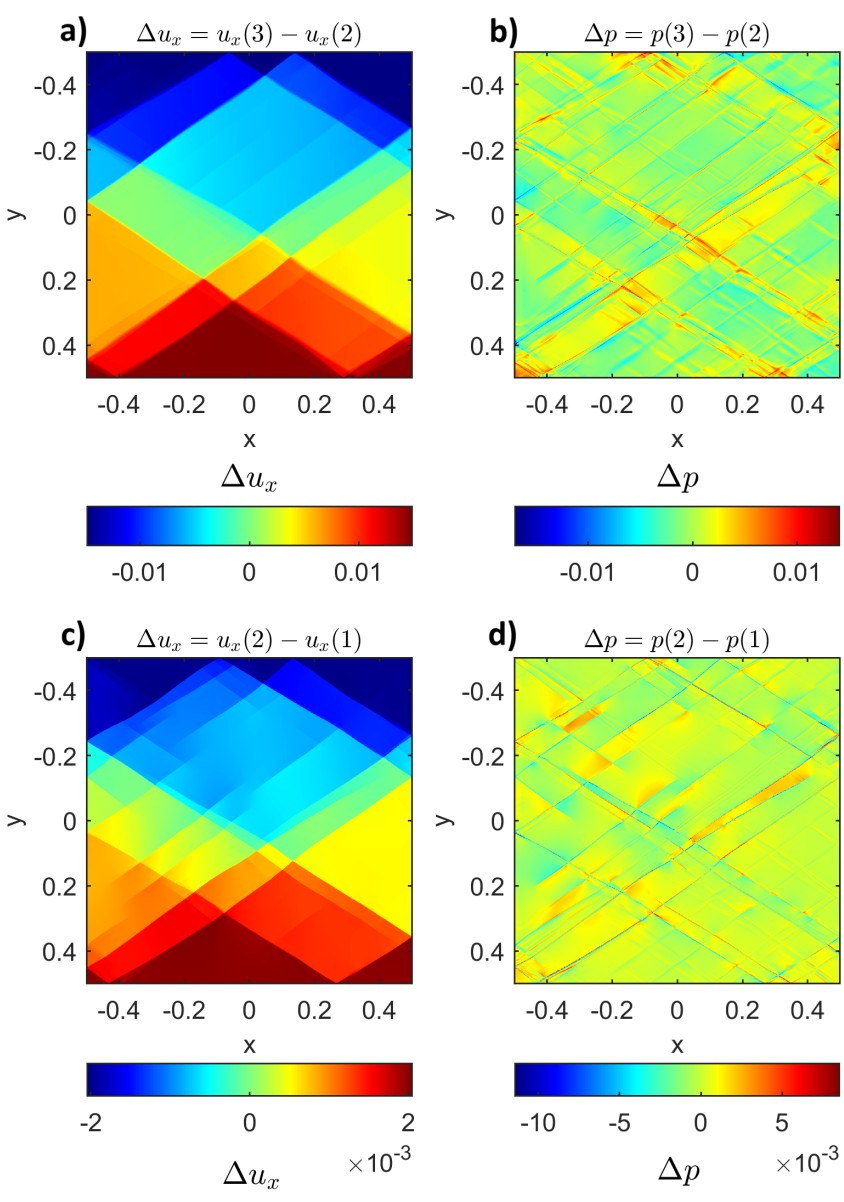

**Figure 15.** Interseismic period and stress drops: numerical simulation of compressible visco-elasto-plastic equations with the resolution of $N = 1023^2$ grid cells for 2500 loading increments in time. Panels (a-b) show displacement increments $\Delta u_x$ corresponding to interseismic periods. Panels (c-d) show displacement increments $\Delta u_x$ corresponding to major stress drops.



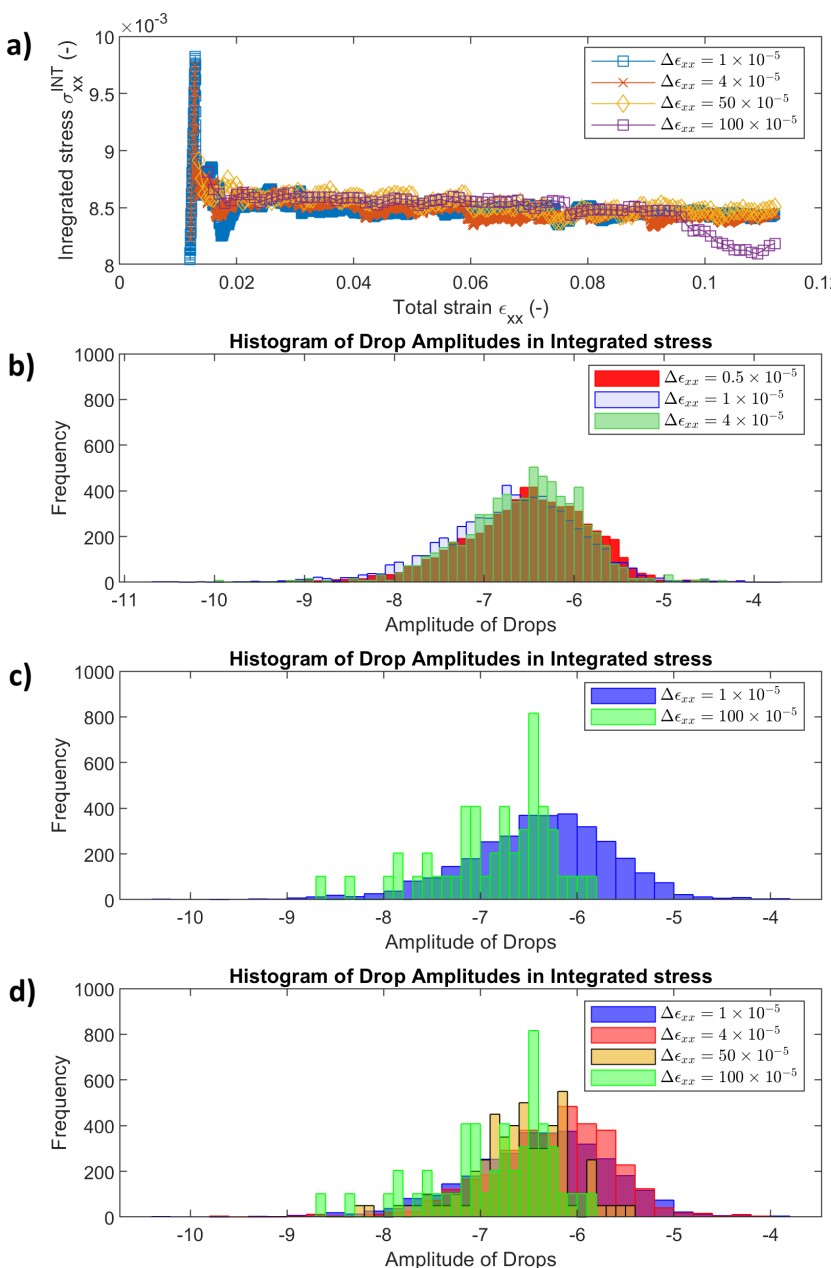

**Figure 16.** Histogram of of stress drop amplitudes for different resolutions. Panel (a) shows the integrated stress $\sigma_{xx}^{\mathrm{INT}}$ versus loading increments. Panel (b) shows the histogram of stress drop amplitudes for three different resolutions. Panel (c) shows the histogram of stress drop amplitudes for two different resolutions. Panel (d) shows the histogram of stress drop amplitudes for four different resolutions.





### 4.9.3 Earthquake nucleation due to a single stress drop

Figure 17 displays the integrated stress $\sigma_{xx}^{\mathrm{INT}}$ versus loading increments and the wave fields (velocity $v_x$ and pressure $p$) at the initial stage (Figures 17a-b) and after 250 physical time steps (Figures 17c-d). The initial wavefield pattern is complex mostly corresponds to nucleation is shear (i.e., as a double-couple mechanism) and in a the volumetric component (pressure). The

365     velocity field exhibits high amplitudes (Figures 17b and 17d), indicating that the shear component has high amplitudes.



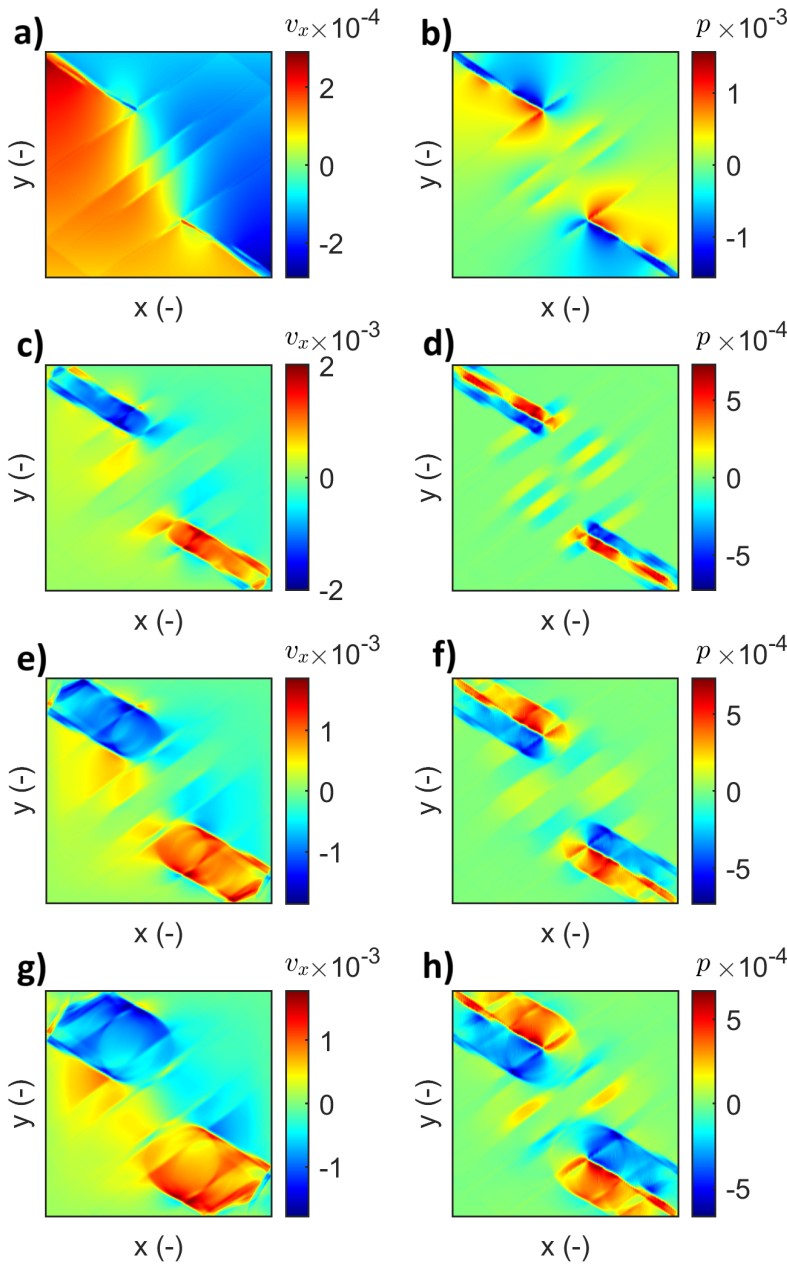

**Figure 17.** Earthquake nucleation due to a single stress drop. Panel (a-b) show the wave fields (velocity $v_x$ and pressure $p$) at the initial stage. Panels (c-d) show the wave fields (velocity $v_x$ and pressure $p$) after 250 time steps. Panels (g-h) show the wave fields (velocity $v_x$ and pressure $p$) after 500 time steps. Panels (c-d) show the wave fields (velocity $v_x$ and pressure $p$) after 750 time steps.





## 5 Discussion

### 5.1 The nature of stress drops

As can be seen in Figure 14, during the loading, many stress drops occur. These stress drops correspond to transitions where the system moves from one quasi-static equilibrium to another due to the inability of strain localization to continue growing in the prescribed direction. Once the local stresses exceed the yield criteria, plastic deformation is activated, causing a redistribution of stresses and, consequently, a rapid drop in stress. This process mimics the mechanics of fault rupture, where the accumulation of strain energy leads to a sudden release in the form of an earthquake.

The observed stress drops are consistent with those expected in elasto-plastic materials, where plastic yielding results in rapid shifts in the stress state. In our simulations, these stress drops are sharp and distinct, especially at higher temporal and spatial resolutions. This behavior reflects the real-world phenomenon of earthquake nucleation, where a sudden stress drop corresponds to a seismic event.

In addition to stress drop magnitudes, the spatial patterns of strain localization play a critical role in determining the nature of stress release. In particular, localized shear bands act as conduits for stress concentration, dictating how and where stress is released. The interplay between elastic loading during the interseismic period and plastic deformation during stress drops provides a simplified but effective model for capturing earthquake cycles.

### 5.2 Role of regularization in elasto-plastic simulations

Regularization is crucial in numerical simulations of elasto-plastic materials, particularly in models involving strain localization. Without regularization, simulations can exhibit unrealistic results, such as the formation of infinitely narrow shear bands. In our study, the regularization parameter $\eta^{\mathrm{vp}}$ was carefully chosen to prevent such artifacts while preserving the physical realism of the stress and strain fields.

The absence of regularization can lead to numerically unstable results, where stress drops occur too frequently or are too abrupt, producing non-physical behaviors in the model. On the other hand, excessive regularization can overly smooth out stress and strain fields, suppressing the formation of localized shear bands and reducing the occurrence of stress drops. The balance between these extremes is key to accurately modeling the dynamic behavior of elasto-plastic systems.

Our results demonstrate that appropriate regularization is necessary to capture both the broad and fine-scale features of earthquake nucleation, such as the spatial distribution of strain localization and the timing and magnitude of stress drops. The findings align with prior studies that show the importance of regularization in stabilizing numerical simulations of plastic deformation while maintaining physical accuracy (Popov and Sobolev, 2008; Duretz et al., 2018).

### 5.3 3D simulations with zero regularization

Alkhimenkov et al. (2023) performed 3D simulations of a single-phase elasto-plastic model with zero regularization. These simulations provide valuable insights into how strain localization and stress drops manifest in fully 3D domains. The con-



vergence tests performed in 3D, both in temporal and spatial resolutions, show good agreement with the results of the 2D simulations presented in this study.

The extension to 3D is important because it allows for a more realistic representation of fault systems, which are inherently three-dimensional in nature. In 3D, stress and strain fields exhibit more complex behaviors, such as the formation of multiple interacting shear bands or the influence of out-of-plane stresses on fault slip. The fact that the 3D simulations without regularization produced results consistent with our 2D study underscores the robustness of the elasto-plastic model used in this research.

### 5.4 Implications for earthquake nucleation and fault mechanics

The results from both 2D and 3D simulations provide important implications for our understanding of earthquake nucleation and fault mechanics. The stress drops observed in our models are analogous to the rapid release of accumulated stress during seismic events, suggesting that elasto-plastic models can effectively capture the mechanics of rupture initiation. The periodic nature of stress drops, interspersed with slower periods of strain accumulation, closely mirrors the earthquake cycle seen in nature.

These findings also highlight the role of strain localization in controlling fault weakening and slip behavior. The formation of shear bands, which localize deformation, is crucial for dictating the location and extent of fault slip during seismic events. This phenomenon is particularly important for understanding the evolution of fault zones, where repeated cycles of strain localization and stress drop shape the mechanical properties of the fault over time.

### 5.5 Comparison to rate-and-state friction models

While our study focuses on elasto-plasticity as the primary mechanism driving stress drops and strain localization, it is important to consider how these results compare to traditional rate-and-state friction models. Rate-and-state friction models have been successful in explaining many aspects of earthquake nucleation and fault slip behavior, particularly through their ability to capture velocity weakening and strengthening behaviors.

In contrast, the elasto-plastic model used in this study does not rely on velocity-dependent friction laws but instead captures stress drops through plastic yielding. This distinction is important because it provides an alternative explanation for how faults might weaken and slip during seismic events. The fact that our elasto-plastic model produces stress drops without needing to invoke velocity weakening suggests that plastic deformation alone may be sufficient to explain certain aspects of fault slip behavior.

### 5.6 Apparent coefficient of friction

A mature fault is one that has experienced significant slip over geological time, leading to the development of fault gouge—a fine-grained, granular material that accumulates within the fault zone. This fault gouge undergoes elasto-plastic deformation,





exhibiting plastic flow when stresses exceed the yield limit. In elasto-plastic models of fault zones, slip is accommodated by a combination of elastic deformation and plastic shear flow, with the latter dominating once the material reaches its plastic limit.

The apparent coefficient of friction, $\mu_a$, is consistently lower than the real internal friction, $\mu$, due to the complex plastic flow
430 within the fault gouge. As described by Byerlee and Savage (1992), $\mu_a$ is related to the internal friction angle $\bar{\varphi}$ by $\mu_a = \sin(\varphi)$, while the real coefficient of friction is given by $\mu = \tan(\bar{\varphi})$.

In the case of the fault gouge material, as illustrated by the Mohr's circle in Figure 4b, the apparent angle $\varphi_A \approx 27°$ results in $\mu_a \approx 0.51$, significantly lower than the real friction value of $\mu = 0.6$. This reduction in the apparent coefficient of friction is a direct consequence of the plastic shear flow within the fault gouge, which reduces the shear stress required for slip. The lower
435 $\mu_a$ facilitates fault slip at reduced shear stresses, thus influencing fault stability and the potential for earthquake initiation. This behavior emphasizes the critical role of plastic deformation in the mechanics of fault weakening during seismic events.

### 5.7 Limitations and future work

While our study provides valuable insights into the mechanics of stress drops and strain localization, it is important to acknowledge the limitations of the current model. First, the model assumes homogeneous material properties, which may oversimplify
440 the complexity of real fault zones. In reality, fault zones are highly heterogeneous, with variations in material properties that can significantly influence fault behavior.

Additionally, the current model does not account for fluid-rock interactions, which are known to play a significant role in fault weakening and earthquake nucleation, particularly in fluid-saturated fault zones. Future work could extend this model to include poroelastic effects or fluid migration, which would provide a more complete picture of the processes governing fault
445 slip.

Finally, while our results are based on 2D simulations, future studies should further explore 3D simulations, which are more representative of real fault systems. Extending the present model to include 3D geometries, along with higher-resolution simulations, would allow for a more detailed investigation of fault mechanics and earthquake nucleation.



## 6   Conclusions

In this study, we investigated stress drops and earthquake nucleation in idealized elasto-plastic media through two-dimensional numerical simulations. Our results underscore the critical role of both temporal and spatial resolutions in capturing the evolution of stress and strain fields during seismic cycles. The convergence tests demonstrated that finer temporal discretization sharpens the observed stress drops and leads to lower minimum stress values, underscoring the importance of accurately resolving dynamic stress changes. Similarly, spatial resolution tests showed that while broad patterns of accumulated strain were consistent

across different resolutions, higher-resolution grids provided significantly more detail, capturing intricate strain localization and stress redistribution mechanisms that are essential for modeling realistic earthquake behavior.

The analysis of interseismic periods and stress drops revealed that displacement gradually accumulates during the interseismic phase, followed by intensified strain during major stress drops. This behavior mirrors the natural earthquake cycle, where periods of slow, aseismic slip are followed by rapid, seismic slip events that release accumulated strain energy. Furthermore,

our detailed investigation of earthquake nucleation due to a single stress drop revealed complex initial wave field patterns, with high-amplitude shear components dominating the response, providing insights into the mechanics of rupture initiation.

One of the key contributions of this study is the demonstration that simple elasto-plastic models, when coupled with high-resolution discretizations, are capable of reproducing key features of earthquake nucleation and stress drop behavior, without relying on more complex frictional laws or velocity-dependent weakening mechanisms. This indicates that plastic yielding

alone can account for some of the fundamental processes governing fault slip and rupture.

Our findings have several important implications for seismic hazard assessment and the development of predictive models. First, they emphasize the necessity of incorporating high-resolution spatial and temporal discretizations into numerical models to accurately capture the localized and transient phenomena that govern earthquake nucleation. Second, the results highlight the role of plastic deformation in fault weakening and rupture, suggesting that plasticity should be considered alongside traditional

frictional models in future earthquake simulations.

Finally, while our study has focused on two-dimensional idealized elasto-plastic media, the insights gained here provide a solid foundation for extending the analysis to more complex, three-dimensional fault systems and heterogeneous materials. Future research could explore the interactions between plasticity, material heterogeneity, and fluid migration, providing a more comprehensive understanding of the mechanics underlying seismic events. By advancing these models, we move closer to

developing more accurate, physics-based tools for predicting earthquake behavior and mitigating seismic risk.

*Code availability.* The software developed and used in the scope of this study is licensed under MIT License. The latest versions of the code is available from a permanent DOI repository (Zenodo) at: https://doi.org/10.5281/zenodo.13942793 (last access: 17 October 2024) (Alkhimenkov et al., 2024b). The repository contains code examples and can be readily used to reproduce the figures of the paper. The codes are written using the Matlab, and CUDA C programming languages. Refer to the repositories' README for additional information.





*Author contributions.* YA designed the original study, developed the codes and algorithms, performed benchmarks, created the figures, and wrote the manuscript. LK contributed to the study design, helped develop the codes and algorithms, and edited the manuscript. YP provided early work on accelerated PT methods, contributed to the study design, helped develop the codes and algorithms, assisted with the interpretation of the results, edited the manuscript, and supervised the work.

*Competing interests.* The contact author has declared that none of the authors has any competing interests.

*Financial support.* Yury Alkhimenkov gratefully acknowledges support from the Swiss National Science Foundation, project number P500PN_206722. Lyudmila Khakimova and Yury Podladchikov thank the Russian Science Foundation (project №24-77-10022) for supporting the development of numerical algorithms and facilitating large-scale GPU-based computations.



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
