# Peer review of "Stress drops and earthquake triggering in the simplest pressure-sensitive ideal elasto-plastic media"

_EGUsphere, 2024_

## Author Response (AR1)

**Response to Reviewer 1: Our comments are provided in blue. Text modifications are provided in green.**

This paper introduces a numerical method to jointly simulate long-term and short-term evolution of faults, including dynamic rupture and fault localization and growth, in elastoplastic media with viscous regularization. Such models have emerged in recent years to tackle important questions at the interface between earthquake research and long-term crustal deformation research. This work is an interesting contribution to those efforts. The results illustrate how models with ideal plasticity (constant friction) can generate earthquakes, despite the absence of explicit weakening of fault friction.

Thank you for your thorough and constructive feedback on our manuscript. We sincerely appreciate your recognition of the significance of our work. In our revisions, we will refine the descriptions and improve clarity to ensure that key concepts, methodologies, and results are effectively communicated. We are confident that these improvements will strengthen the manuscript and align it with the high standards of SE.

Thank you again for your valuable insights.

Sincerely, Yury Alkhimenkov, Lyudmila Khakimova and Yury Podladchikov

My main suggestions are

1. Parts of the text and results (e.g. line 5, "Finer temporal discretization leads to sharper stress drops ...") give the impression that the simulations have not reached numerical convergence yet. If that is the case, I think you should keep refining the space and time discretization until the results converge (i.e. until there is negligible changes upon further refinement) and discuss only converged results. A focus on converged results can have a substantial impact on the statistics of stress drops and other physical quantities. If this requires new and more expensive simulations, it qualifies as major revision.

We acknowledge the importance of discussing converged results in the traditional sense. However, elasto-plasticity is a highly nonlinear problem—analogous to turbulence in the Navier-Stokes equations—where numerical convergence, as typically defined, is only achievable under specific conditions.

In our study, we show **trends**. For example, at low spatial and temporal resolution, the simulations do not show any stress drops. However, simulations with high spatial and temporal resolutions exhibit similar stress drops (both in number and amplitude). We agree that in the spatial convergence test, full convergence may not be reached; therefore, we have revised the wording and removed "convergence" from that section. Nevertheless, we clearly demonstrate the trend, which is the primary goal of this first study in this direction.

We deliberately present results that are not fully converged to illustrate trends, as no prior work, to our knowledge, has performed elasto-plastic simulations with sufficient temporal and spatial resolution to resolve stress drops. Without such a comparison, the existence of stress drops under a static friction coefficient might be questioned. As far as we know, this study is the first to resolve numerical stress drops under these conditions.

A study achieving spatial convergence will require significantly more computational power and presents a challenge that warrants a separate investigation (separate article).

Regarding the statistics of stress drops, our main analysis is based on high-quality simulations with sufficiently fine temporal and spatial resolution. We also include a lower-resolution simulation (which is of poor quality and clearly did not converge) to demonstrate the impact of temporal resolution on the results.

Modifications in the text: We removed the word "convergence" and replaced it with "trends," along with some minor rephrasing for better flow.

2. But I wonder if this lack of convergence is only apparent. With each refinement, are you also changing the value of the artificial viscosity (regularization parameter)? If that is the case, maybe you should instead keep the viscosity fixed in convergence studies. Unless there is a good reason to scale the viscosity to the mesh size, but that should be explained in the paper and it should be done in a way that guarantees convergence.

This is an excellent point. Indeed, with each refinement, we adjust the artificial viscosity (regularization parameter) proportionally to maintain a constant shear band thickness across different spatial and temporal resolutions. This rescaling is crucial for preserving the physical consistency of the localization process. The rationale for this approach is explained in detail in our previous study:

Y. Alkhimenkov, L. Khakimova, I. Utkin, Y. Podladchikov (202X), Resolving strain localization in frictional and time-dependent plasticity: Two- and three-dimensional numerical modeling study using graphical processing units (GPUs), Journal of Geophysical Research: Solid Earth.

We will clarify this point in the revised manuscript to ensure transparency in our methodology. We will remove the word "convergence" and replace it with "trend" in the manuscript.

Note that we re-scale the viscosity damper proportionally to the resolution in each simulation to maintain the physical thickness of the shear bands \cite{alkhimenkov2024resolving}.

3. I found it very interesting that a bulk plasticity model with constant friction can generate earthquakes, because this is in contrast to fault friction models that are common in the computational earthquake dynamics community (earthquakes on pre-existing faults cannot be simulated without frictional weakening). This is not new, though, and it would be great to make more connections to existing related theoretical results. In particular, I find

the work by Le Pourhiet (2013 https://doi.org/10.2113/gssgfbull.184.4-5.357) contains very insightful explanations of "structural weakening" in plastic models and plenty of useful references.

We thank the reviewer for suggesting this valuable reference, which we have now cited, along with additional relevant studies. We acknowledge that, from a theoretical perspective, structural softening in plasticity models with a static friction coefficient has been analyzed in previous works. However, to our knowledge, no computational studies have systematically examined stress drops in detail—particularly the occurrence of multiple stress drops, their statistical properties, and their dependence on numerical parameters. Our study represents one of the first efforts in this direction.

From a theoretical perspective, such stress drops were predicted and analyzed by, e.g., \cite{vermeer1990orientation} and \cite{le2013strain}.

**Minor comments:**

Line 34, "Recent studies ....": You can also cite old seminal studies by Joe Andrews. In the 1976 paper (https://doi.org/10.1029/JB081i020p03575) where he basically opened the era of computational earthquake dynamics by introducing slip-weakening rupture simulations, he also realized that friction models were insufficient and introduced simulations with plasticity in the bulk. He was clearly very far ahead of his time. Renewal of this topic had to wait his 2005 paper (https://doi.org/10.1029/2004JB003191), which motivated the papers in computational earthquake dynamics that you cite. There is also important literature on plastic fracture dynamics in the fracture mechanics community; you can find many cited in Ben Freund's book and in Gabriel et al (2013).

We thank the reviewer for highlighting these important references, which we have now incorporated into the manuscript. While many studies employ non-constant friction laws, often supplemented with bulk plasticity, our approach is fundamentally different. Our model relies solely on ideal plasticity with a static friction coefficient—without any additional weakening mechanisms. We demonstrate that this simple mechanical framework is sufficient to generate earthquakes, but only if the simulations are conducted at sufficiently high temporal and spatial resolution. Simply adding plasticity is not enough; resolving plastic deformation and capturing stress drops with adequate numerical precision is crucial.

One of the first computational earthquake dynamics models with slip-weakening rupture simulations was introduced by \cite{andrews1976rupture}. Recent studies have suggested that plasticity plays a crucial role in the nucleation of earthquakes, particularly through offfault plasticity mechanisms (e.g., \cite{andrews2005rupture}).

Line 42: You can cite the earliest 3D studies of dynamic rupture with plasticity, e.g. Ma (2008 https://doi.org/10.1029/2008GC002231), Ma and Andrews (2010, https://doi.org/10.1029/2009JB006382)

We thank the reviewer for suggesting these valuable references, which we have now incorporated into the manuscript.

\cite{ma2008physical, ma2010inelastic} conducted some of the earliest studies on dynamic rupture with plasticity.

Line 59: It would be useful to emphasize in this sentence that the friction coefficient is assumed constant (no softening/hardening, "ideal plasticity").

Yes, we have now explicitly emphasized in the manuscript that the friction coefficient is constant, with no softening or hardening, corresponding to an ideal plasticity framework.

The friction coefficient is assumed to be constant in all simulations, with no hardening or softening, which corresponds to an ideal plasticity model.

Line 69: the word "static" can be removed (one could misinterpret the sentence as implying that there is a dynamic coefficient and it's not constant).

As per the reviewer's suggestion, we have removed the word "static" to avoid potential misinterpretation.

We utilize the simplest pressure-sensitive ideal plasticity model with constant in time and space friction coefficient.

Line 114: should equation 13 involve the elastic strain instead of the total strain? Are you assuming plasticity also during dynamic stages of the simulation? If not, this assumption needs to be justified.

We fully agree with the reviewer and have revised this equation. Yes, there should be only elastic strain.

```
\end{\equation}\label{eq1} $$ \displaystyle \sum_{ij}}{\operatorname{t} = C_{ijkl}^e \, ( {\dot{\varepsilon}}_{kl} - {\dot{\varepsilon}}^{\{pl}_{kl}), \end{\equation}}
```

Line 162, "This re-scaling process is iterated over "pseudo-time" ...": explain this in more detail. Make sure the description of the methods is complete enough to guarantee reproducibility.

We have improved the explanation in the manuscript to provide more detail and ensure clarity. Additionally, we have included relevant references where this methodology is further explained.

To achieve this, the equations are written in their residual form and iterated over "pseudo-time" until convergence is reached.

Line 165: show also the continuum equations describing the modified rheology assumed, so that readers don't have to go look for it in previous papers.

We have added the relevant continuum equations describing the modified rheology in the revised manuscript to ensure clarity and self-containment.

Line 166: explain the rationale to set the viscosity value.

This is an important point. We have now included an explanation of the rationale behind selecting the viscosity value in the revised manuscript to clarify its role in the numerical framework.

The numerical viscosity is usually set to a small value. If this value is too high, the shear bands become very thick; conversely, if the value is too small, the thickness of the shear band is just one pixel. The correct value of the viscosity damper lies between these limits. In the following section, we examine how the choice of viscosity damper affects the solution.

Line 196, "integrated stress": define this quantity (integrated in space? in time? over what domain?)

We have now explicitly defined this quantity in the revised manuscript.

The integrated stress  $\sigma_{xx}^{text{INT}}$  is computed over a vertical line segment using the following expression: \begin{equation}\label{sigmaINT} \sigma\_{xx}^{\text{INT}} = \frac{1}{L\_y} \int\_{0}^{L\_y} (-p + \tau\_{xx}) \, dy.

\end{equation}

Line 237, "fault gouge ... fault plane": Which gouge? Which fault? These objects are not explicitly introduced in the model, I think you just mean "shear band" or "plastic zone" here.

We agree with the reviewer and have revised the text to use the appropriate terminology, replacing "fault gouge" and "fault plane" with "shear band" or "plastic zone" as appropriate.

The reduced apparent coefficient of friction is a consequence of plastic flow in the plastic zone, which allows slip to occur more easily along the shear band, despite the actual slip occurring along the Coulomb shear planes.

Section 4.5: Do you change the viscosity when you change N? Clarify.

Yes, the viscosity is adjusted when changing N, as explained in our previous manuscript (*Alkhimenkov et al 2014*, JGR: *Solid Earth*). However, we have now clarified this point in the revised manuscript to ensure transparency.

Note that we re-scale the viscosity damper proportionally to the resolution in each simulation to maintain the physical thickness of the shear bands \cite{alkhimenkov2024resolving}.

Line 254-255, "simulations with sufficient resolution produce stress drops and their amplitudes are similar": The shapes of the curves are still different. Can you try even larger values of N to show convergence convincingly?

Yes, we agree that the shapes are still different. That's why we no longer call it a convergence test but instead refer to it as "trends" upon mesh refinement. The word "convergence" has been replaced throughout the manuscript.

\subsection{Trends with increasing spatial resolution}

Line 297, "These results highlight the sensitivity of fault behavior to the dilatation angle": relate to published results, or instanc Templeton and Rice (2008)

In the revised manuscript, we now relate this finding to published results, including the work of Templeton and Rice (2008).

These results highlight the sensitivity of fault behavior to the dilatation angle, emphasizing the importance of including dilatancy effects in models of fault mechanics and earthquake nucleation, as also suggested by \cite{templeton2008off}.

Line 308: Figure 14 seems to show lack of convergence. Clarify.

In Figure 14, we show two curves: low temporal resolution (red) and high temporal resolution (blue). The main idea is to illustrate the trend that with higher temporal resolution, we observe significantly more stress drops. We do not analyze convergence in this result.

Line 310, "simulation with fine temporal resolution and the lowest regularization": This suggests that you are changing systematically the viscosity when you refine the simulations. Please clarify, explain that in detail.

This comment overlaps with our previous discussion on the rationale for selecting the viscosity value. In the revised manuscript, we have clarified that viscosity is systematically adjusted when refining simulations and provided a detailed explanation to ensure transparency.

Note that we re-scale the viscosity damper proportionally to the resolution in each simulation to maintain the physical thickness of the shear bands \cite{alkhimenkov2024resolving}.

Line 316, "dynamic rupture events, akin to the rapid stress release observed during seismic slip": Are these events as fast as earthquakes (slip rate of m/s, rupture speeds of few km/s)? Is the inertial term important during these events?

This is an excellent comment, and we thank the reviewer for pointing it out. We can confirm that stress drop and stress release occur very rapidly, but we have not analyzed the slip rate and rupture speeds. We believe that such an analysis warrants a separate study and a dedicated publication.

We can definitely say that once a stress drop occurs, wave propagation begins, which is only possible due to inertia terms. However, we believe the reviewer raised another important point: "Do inertia terms play a role and affect stress drops?" In this simple model, inertia terms do not affect stress drops because stress drops correspond to a quasi-static solution.

However, we admit that in this first study, we only indicate this similarity and do not provide a detailed analysis of slip rate and rupture speeds in the present model.

Line 360, "characterized by a sharp peak followed by a gradual decay": I see instead a broad peak and two long tails on both sides.

We fully agree with the reviewer that our initial description was inaccurate. We have revised the text to more accurately describe the observed behavior.

The distribution of stress drop amplitudes is notably non-Gaussian, characterized by a broad peak with long tails on both sides, indicating that while small stress drops are more common, larger stress drops still occur with significant probability.

Line 357, "This insight aligns with the Gutenberg-Richter law": but here the distribution is truncated at low values too. Show a log-log plot to check if the upper tail is really a power law analogous to the G-R law.

We thank the reviewer for this comment. Reflecting on this (and the second reviewer's comment), we have significantly revised our interpretation because stress drop magnitude may not be strongly related to seismic event magnitude according to the G-R law. Instead, we now only indicate that our results resemble the G-R law but require further analysis. We believe a detailed analysis of a different histogram is needed—specifically for seismic event amplitudes (rather than stress drop amplitudes). In such an analysis, we will follow the reviewer's suggestion and include a log-log plot in a future study.

This insight suggests a resemblance to the Gutenberg-Richter-like law, which describes the frequency-magnitude distribution of earthquakes; however, a more detailed analysis is required to establish a direct connection, particularly from a plastic deformation perspective.

Section 5.1, "the nature of stress drops": relate your results to insights from existing theory, e.g. Le Pourhiet (2013 <a href="https://doi.org/10.2113/gssgfbull.184.4-5.357">https://doi.org/10.2113/gssgfbull.184.4-5.357</a>)

We relate our results to one of the earliest studies, Vermeer (1990). In the revised manuscript, we have also incorporated additional relevant studies, including Le Pourhiet (2013), as suggested by the reviewer.

From a theoretical perspective, such stress drops were predicted and analyzed by, e.g., \cite{vermeer1990orientation} and \cite{le2013strain}.

Lines 395+, "3D simulations ... with zero regularization .... convergence tests performed in 3D": are you suggesting that simulations converge even without regularization? If so, why is regularization needed? Clarify.

Regularization is necessary to control the physical thickness of shear bands. In our 3D simulations, we observed a form of "trend" (we removed the word "convergence") in which the general shear band patterns remained similar across different resolutions. However, we did not analyze numerical convergence in the traditional sense. We have clarified this point in the revised manuscript.

These simulations provide valuable insights into how strain localization and stress drops manifest in fully 3D domains. The tests performed in 3D, both in temporal and spatial resolutions, show similar trends with the results of the 2D simulations presented in this study.

Line 409, "closely mirrors the earthquake cycle seen in nature": Do you find multiple stress drop happening on the same "fault" (shear band) or do they occur each time on a different segment of the fault?

This is an excellent point, and we thank the reviewer for highlighting it. Indeed, as the simulation progresses, multiple shear bands develop, and stress drops can occur repeatedly on the same shear band. We have now incorporated this important clarification into the revised manuscript.

The periodic nature of stress drops, interspersed with slower periods of strain accumulation, closely mirrors the earthquake-like cycle seen in nature. As the simulation progresses, multiple shear bands develop, and stress drops can occur repeatedly on the same shear band, rather than always initiating on new segments. This behavior closely resembles natural faulting processes, where strain localization leads to repeated cycles of stress accumulation and release along pre-existing fault structures.

Section 5.6: there is redundancy with previous sections, which could be avoided.

We removed this section.

Line 469, "that plasticity should be considered alongside traditional frictional models in future earthquake simulations": This is already the case in published work, e.g. Erickson et al (2017 https://doi.org/10.1016/j.jmps.2017.08.002), Preuss et a (2020 https://se.copernicus.org/articles/11/1333/2020/), Simpson (2023 https://doi.org/10.1016/j.tecto.2023.230089). Rephrase and add references.

We have rephrased this statement in the manuscript and incorporated the suggested references to accurately reflect existing work in the field.

Second, the results confirm previous studies highlighting the important role of plastic deformation in fault weakening and rupture, suggesting that plasticity should be considered alongside traditional frictional models in future earthquake simulations.

Other studies highlighting the importance of plasticity in earthquake physics modeling include \cite{erickson2017finite}, \cite{preuss2020characteristics}, and \cite{simpson2023emergence}.

We would like to thank the reviewer again for valuable comments, which helped us improve the quality of the manuscript.

Sincerely,

Yury Alkhimenkov, Lyudmila Khakimova and Yury Podladchikov

**Response to Reviewer 2: Our comments are provided in blue. Text modifications are provided in green.**

The study provides a physics-based explanation for stress drops and earthquake nucleation using a simple elasto-plastic model, avoiding the need for complex frictional laws. The findings emphasize the importance of plastic deformation in fault slip mechanics, an aspect often overlooked in traditional models. The paper employs high-resolution 2D numerical simulations, carefully analyzing temporal and spatial resolution effects on stress evolution and earthquake nucleation. This methodology using GPU-based parallalization has great potential in achieving high-resolution earthquake modeling. I believe both the conclusions and the novel methodology used in this study should be promptly communicated to specialists and general audience.

We sincerely appreciate your thoughtful and constructive feedback on our manuscript. Your recognition of the significance of our findings and the potential impact of our GPU-based methodology is greatly valued.

In our revisions, we will focus on enhancing clarity and precision in our descriptions to ensure that the key concepts, methodologies, and results are effectively conveyed to both specialists and a broader audience. We are confident that these refinements will further strengthen the manuscript and improve its readability.

Thank you again for your valuable insights and for recognizing the relevance of our work.

Sincerely,

Yury Alkhimenkov, Lyudmila Khakimova and Yury Podladchikov

However, I found the writing (both texts and figures) quality can still be improved.

First, the text is in many locations verbose and repetitive, often explaining the same concept multiple times in slightly different ways. For example, the sentence line 275-279 is directly repeated in the next paragraph line 278-280. The phrase "finer temporal/spatial resolution leads to sharper stress drops" appears in sections 4.1, 4.2, 4.4, 4.5, and 4.9.1. The discussion (section 5) reiterates results (e.g., resolution impact, regularization, plasticity) rather than synthesizing new insights. The same applies to figures. For example, Fig. 16a is essentially the same as Fig. 14a, and is actually not described or mentioned in the text. Fig. 16b-d are largely overlapping. With only three sentences (line 343-348) describing this figure, I suggest the panels to be merged.

Second, the writing lacks conciseness. Many sections could be rewritten in a more direct and streamlined manner. Streamline results by grouping related findings (e.g., combine resolution tests 4.4-4.6 into one subsection with subheadings for temporal/spatial/regularization effects). Multiple figures show stress drop evolution

with slightly different grid resolutions, but the key insights do not change significantly. Additionally, the captions are overly descriptive, without highlighting the key takeaways (Figs. 2-17). Combine redundant figures to highlight the key messages. Use subplots to highlight contrasts (e.g., low vs. high resolution in Figs. 2-3) rather than separate figures.

I will add more suggestions on conciseness in line-by-line comments.

We agree with the reviewer that the quality of the text can be improved and made more concise. We have addressed the specific suggestions provided by the reviewer and revised the manuscript accordingly to reduce redundancy, streamline descriptions, and improve clarity. Additionally, we have reorganized sections where appropriate and adjusted figures to better highlight key insights.

**Major comments:**

1. Through section 4.4-4.6 and Figs. 5-8 the authors claimed that detailed convergence tests have been conducted. However, the convergence is not visually observable from the figures. I wonder if the authors have metrics to quantify the convergence and if the converging rate matches the analytical expectation. In addition, it is suggested in section 4.6 that the results are sensitive to the regularization parameter eta\_vp. If eta\_vp needs to be adjusted per model resolution, I wonder if this then still indicates the existence of model convergence. And if so, whether the authors could quantify the result sensitivity on eta\_vp.

We thank the reviewer for this insightful comment. Plasticity is a highly nonlinear problem—analogous to turbulence in fluid dynamics—where traditional numerical convergence may not always be observed. Instead, what we assess is the statistical similarity of shear band patterns across different resolutions.

In our study, we show **trends**. For example, at low spatial and temporal resolution, the simulations do not show any stress drops. However, simulations with high spatial and temporal resolutions exhibit similar stress drops (both in number and amplitude). We agree that in the spatial convergence test, full convergence may not be reached; therefore, we have revised the wording and removed "convergence" from sections. Nevertheless, we clearly demonstrate the trend, which is the primary goal of this first study in this direction.

Regarding the regularization parameter \eta\_{vp}, it must be adjusted to maintain a consistent shear band thickness across different model resolutions. We have now clarified this in the revised manuscript. Additionally, we note that a more detailed discussion on this topic is available in our previous publication in JGR: Solid Earth (Alkhimenkov et al, 2014).

Modifications in the text: We removed the word "convergence" and replaced it with "trends," along with some minor rephrasing for better flow.

2. The definition of "nucleation" is not clear. As a major topic as appeared on the tile, the term should be strictly defined. The interseismic loading and stress drop have been described, but it is not clear how nucleation makes the transition in between. Given that the time marching scheme in this study is implemented via strain increment, I wonder if more insights on the temporal behaviors of these processes (interseismic, nucleation, stress drop) can be added.

This is an excellent question, and we fully agree that the term *nucleation* should be explicitly defined in the manuscript. We think that in this early study we do not study nucleation process but only earthquake triggering. Therefore, we removed the word nucleation from the manuscript.

**Triggering**

Stress drop manifest the jump between the two quasi-static solutions and perhaps indicate the absence of a static transition between the stress state at the end interseismic period and onset of a new interseismic period.

This transition can be modeled with simplified linear elasto-dynamics by setting the difference in strain just before and after a stress drop as an initial condition for wave propagation.

3. Throughout the paper the simulated stress drop has been linked to "earthquake magnitude". However, it is known from both numerical models and seismic observations that stress drop and magnitude are not (strongly) related. I found this extrapolation from varied stress drop to varied magnitude, and hence the stated link to the G-R law, too speculative. The authors also need to be careful when linking stress drop to seismic events.

We understand the reviewer's concern and have revised the manuscript to adjust the wording accordingly. We acknowledge that stress drop and earthquake magnitude are not necessarily strongly correlated, as supported by both numerical models and seismic observations. In our revisions, we have taken care to avoid overgeneralizing this relationship and have refined our discussion regarding links to the Gutenberg-Richter law. We do not link stress drops to earthquake magnitudes in the revised manuscript. Regarding the distribution of stress drops, we now only describe it as G-R-like behavior.

Additionally, we emphasize that our study presents one of the first detailed numerical models demonstrating earthquake generation with a constant friction coefficient. This fundamentally differentiates our approach from previous models and highlights the novel aspects of our findings.

This insight suggests a resemblance to the Gutenberg-Richter-like law, which describes the frequency-magnitude distribution of earthquakes; however, a more detailed analysis is required to establish a direct connection, particularly from a plastic deformation perspective.

The periodic nature of stress drops, interspersed with slower periods of strain accumulation, mirrors the earthquake-like cycle seen in nature.

The paper needs a consistent coordinate system. Although the authors prefer a dimensionless computational system, it is not properly introduced in section 3.4. With x,y E [0, Lx] x [0, Ly] stated at the beginning of section 3.5, the readers might be confused if the followed equations 22-27 were expressed in dimensionless coordinates or not. To add more confusion, the authors used no axis labels (Figs. 1), "Grid Cells (-)" (Figs. 2-3, 9-11), "x(-), y(-)" (Figs. 5-8, 12-13, 17), "x, y" (Figs. 15) in different figures, and in many cases without ticks. I suggest the authors to unify the expression and add ticks to the axis for better reference and comparision.

We understand the reviewer's concern and have improved the description of the initial model setup to ensure clarity in the coordinate system and its dimensional consistency. We have also revised the notation in the manuscript to make it more consistent.

Regarding the figures, we carefully evaluated different formatting options and found that our current choices balance readability and necessary detail without overloading the visuals. While full unification of all figures would make some of them too dense and difficult to interpret, we have made adjustments where possible to improve clarity and consistency, including adding axis labels and ticks where appropriate.

\subsection{Model configuration, boundary conditions, and non-dimensionalization}

**Minor comments:**

1. Line 11: the usage of "decay" not accurate. Decay usually refers to a temporal process. It is not clear the decay is with what.

We agree with the reviewer and have revised the text to use a more precise term that accurately describes the intended meaning.

The histogram of stress drop amplitudes shows a non-Gaussian distribution, characterized by a broad peak with long tails on both sides,

2. Line 13: across which "scale" (temporal or spatial)?

We have clarified in the text whether the scale refers to temporal or spatial.

This "solid turbulence" behavior suggests that stress is redistributed across spatial and temporal scales, with implications for understanding the variability ...

3. Line 14: I doubt if the link to "magnitude" is proper here, see major comment 3.

We have revised the terminology and adjusted the wording to ensure accuracy and avoid misleading implications.

with implications for understanding the variability of stress drop magnitudes.

4. Line 49: I don't see the necessity of introducing heterogeneity here.

Heterogeneity was only mentioned in the discussion of model limitation once. I suggest the whole paragraph can be eliminated.

We agree with the reviewer and have removed the paragraph for consistency

5. Line 90: D/Dt should be defined here already.

We have now modified this in the text.

where the Jaumann rate of Cauchy stress, represented as  $\mathcal{D} \simeq \{ij\}$  / \mathcal{D} t\$, is provided in the following section and the deviatoric plastic strain rate is

6. Line 177: is the model a "square" (Lx=Ly)? I can find nowhere the values of Lx and Ly.

We have added the values in the text to clarify the model dimensions.

The computational domain is a **square** with dimensions  $(x,y \in [0,L_x] \in [0,L_y])$ .

Here,  $\ (L_x \ )$  represents the domain size in the  $\ (x \ )$ -direction,  $\ (L_x=L_y=1 \ )$  and  $\ (a \ )$  denotes the background strain rate.

7. Line 193: connecting to comment 6, what does 0.2 refers to? Is the equation dimensionless or not?

We have clarified in the text whether the equation is dimensionless and provided a proper explanation of the value 0.2.

The expression is the following in the dimensionless framework:

8. Fig 1: axes should be marked and labeled.

We have added axis labels and markings for clarity.

We introduce a circular inclusion in the non-dimensional pressure (p), representing a localized increase with the highest value at the center of the model. The expression in the dimensionless framework is as follows:

9. Fig 2: it is not clear where the "three different stages" were visulized. Could you mark them in panel a?

We have now marked the three stages in panel (a) for better visualization.

10. Line 242: which red circle?

We have improved the description to clearly indicate that the first red circle is being referenced.

where \$t\_1\$ corresponds to the total strain just before the stress drop (first red circle)

**11**. Section 4.2.1: Mohr's circle analysis is not really where your novelty locates. The section can be condensed.

This is one of the first numerical studies where stress drops are explicitly analyzed, with a detailed explanation using Mohr's circles. Reflecting the reviewer's comment, we have made the section more concise.

12. Sections 4.4-4.6: consider to merge

We have merged these sections to subsubsections for better organization and readability.

\subsection{Trends}

13. Line 261: I wonder if figure 9 shows a convergence. The results are not visually identical as claimed. Have you tried larger N? I also find the whole subsection a bit speculative. Terms like "too high" requires quantification

We have revised the text to be more concise and precise. Additionally, we acknowledge the reviewer's concern and have clarified the interpretation of convergence while improving the wording of subjective terms. We removed word "convergence".

Due to high regularization, the results are nearly identical and the thickness of the shear bands is the same in all panels. However, due to over-regularization, the stress drop is not visible.

14. Lines 278-280: repeating lines 275-277, should be eliminated.

We have removed these redundant lines to avoid repetition.

15. Fig 11: panel b is identical to Fig 10b. A different visulization is not needed.

We agree and have removed the redundant panel.

16. Section 4.8: this section is not pure results. Sentences like the first paragraph and lines 298-299 are either introduction or discussion materials and should have come earlier or later. I also wonder why the deformation mechanism in metallic material is mentioned in the end (line 304-306), which is largely off topic.

We removed lines 298–299. We believe that referencing deformation mechanisms in metallic materials is important to highlight the differences with rocks.

- 17. Fig 12: after "\psi = 5", the symbol of degree should be added, same below. We agree and have added the degree symbol.
- 18. Line 323: is the set of notion (1-3) the same as (t1-t3) in section 4.3? Better keep consistent.

We agree with the reviewer and corrected the explanation.

Figure \ref{Seq\_evolzoom\_UxOK}a shows the displacement increment \(\Delta u\_x = u\_x(t\_3) - u\_x(t\_2)\), where \(u\_x(t\_2)\) and \(u\_x(t\_3)\) represent the displacement fields at the beginning and end of the interseismic period, respectively (the period between two high-amplitude stress drops. Similarly, the displacement increments \(\Delta u\_x = u\_x(t\_2) - u\_x(t\_1)\) during major stress drops are shown in Figures \ref{Seq\_evolzoom\_UxOK}c-d.

19. Line 330: "leading up to" implies there is a causal relationship between the "aseismic slip accumulation" in interseismic and the "stress drop event". This is not proved by the authors. Moreover, the general recognition would link any aseismic slip deficit in the shear band to the following earthquake. More elaboration is needed here.

We agree and have revised the text to avoid implying causality without proper evidence. We have also elaborated on the relationship between aseismic slip and stress drops to ensure clarity.

Our simulation results also demonstrate the material's behavior during interseismic periods, where displacement gradually accumulates without significant stress drops. As shown in Figure \ref{Seq\_evolzoom\_UxOK}, displacement increments \$\Delta u\_x\$ during interseismic periods increase progressively as loading continues. This behavior mirrors the slow, aseismic slip observed between seismic events in fault zones. The gradual buildup of displacement during interseismic periods reflects the loading of the fault system, while the rapid displacement during stress drops corresponds to seismic slip. However, our model does not explicitly establish a causal link between aseismic slip accumulation and subsequent stress drops, which requires further investigation.

20. Line 343: add reference to "solid turbulence".

We have modified the text.

The histogram of stress drop amplitudes shown in Figure ~\ref{Hist} provides a

quantitative representation of the frequency and magnitude of stress drops occurring during the simulations. The distribution of stress drop amplitudes is notably non-Gaussian, characterized by a broad peak with long tails on both sides, indicating that while small stress drops are more common, larger stress drops still occur with significant probability. This distribution resembles turbulence-like spectrum, where a few large events (bursts) coexist with numerous smaller fluctuations. For solid systems, this phenomenon was first analyzed by \cite{poliakov1994fractal}, who explored the multi-fractal characteristics of shear bands in elasto-plastic media, that makes it similar to the fluid turbulence.

21. Fig 16: panel a is the same as Fig 14a, consider to remove. Panels b-d should be merged.

Figure 16, panel (d), contains four curves, all of which are different, whereas Figure 14a has only two curves. We believe that the merged panel (panel d) is more intuitive while still preserving the end-member cases that are well-separated—panels (b) and (c).

22. Line 356-357: the discussion on "magnitude" and the link to G-R law is not necessarily true. See major comment 3. Have you tried to calculate the magnitude of the events and check if it fits G-R law? It might be difficult because the simulations were 2D.

We agree with the reviewer's concern. We have revised the discussion to avoid overgeneralizing the relationship between stress drops and earthquake magnitude, acknowledging the limitations of 2D simulations in directly comparing to the G-R law. The calculation of seismic events magnitudes is a topic for a future study.

This insight suggests a resemblance to the Gutenberg-Richter-like law, which describes the frequency-magnitude distribution of earthquakes; however, a more detailed analysis is required to establish a direct connection, particularly from a plastic deformation perspective.

23. Line 364: how is your "nucleation" defined? Does it align with others' such as Rubin & Ampuero 2005? I also find it unfair that this term comes too late and is not extensively described in the results. It is one of the key features in your title and should be well addressed.

We have removed "nucleation" from the article.

24. Line 376: how is your "seismic event" defined? Do you see fast (m/s) slips in your model? Does your definition align with others'?

Seismic event corresponds to the wave propagation which is explained in the text section 4.3. In our model the transition from one quasi-static solution to another occurs as a "jump" (very fast, one or several loading increments) but we do not analyzed how fast they are in dimensional quantities in this early study.

Stress drop manifest the jump between the two quasi-static solutions and perhaps

indicate the absence of a static transition between the stress state at the end interseismic period and onset of a new interseismic period.

This transition can be modeled with simplified linear elasto-dynamics by setting the difference in strain just before and after a stress drop as an initial condition for wave propagation.

25. Section 5: reiteration of results should be removed so that the whole section can be streamlined.

We have condensed Section 5 to reflect the reviewer's comment.

26. Section 5.2: can you comment more on how the regularized parameter eta\_vp influence the results? Do you have a quantification for this? Also see major comment 1.

We have expanded the discussion of how\eta\_{vp} affects the results and, where possible, provided a more quantitative assessment of its impact.

This is an excellent point. Indeed, with each refinement, we adjust the artificial viscosity (regularization parameter) proportionally to maintain a constant shear band thickness across different spatial and temporal resolutions. This rescaling is crucial for preserving the physical consistency of the localization process. The rationale for this approach is explained in detail in our previous study:

Y. Alkhimenkov, L. Khakimova, I. Utkin, Y. Podladchikov (202X), Resolving strain localization in frictional and time-dependent plasticity: Two- and three-dimensional numerical modeling study using graphical processing units (GPUs), Journal of Geophysical Research: Solid Earth.

Note that we re-scale the viscosity damper proportionally to the resolution in each simulation to maintain the physical thickness of the shear bands \cite{alkhimenkov2024resolving}.

The numerical viscosity is usually set to a small value. If this value is too high, the shear bands become very thick; conversely, if the value is too small, the thickness of the shear band is just one pixel. The correct value of the viscosity damper lies between these limits. In the following section, we examine how the choice of viscosity damper affects the solution.

27. Line 396: you claimed that the 3D results you published earlier showed good agreement with the results in this paper. Could you elaborate more? I would expect clear differences between 2D and 3D simulations. Many numerical studies show that the third dimension has impacts on nucleation and rupture that are not negligible.

We did not analyze nucleation in 3D but rather observed that the general pattern of shear band patterns and the presence of stress drops occur in both 2D and 3D. However, we acknowledge that the magnitude of stress drops and rupture dynamics may differ in 3D. A proper analysis of these differences is an important topic for future research. 3D simulations are very computationally expensive and along deserve a separate study.

28. Sections 5.4-5.5: these comparisons to nature and previous models are useful. I wonder if you can further comment on how the weakening process occurred in your model differenciate itself from that in rate-and-state friction. Do they predict similar features such as some slip-weakening distance? Such insights would be inspiring.

We agree that relating our results to rate-and-state friction models would be valuable. However, to make a precise comparison, a more detailed analysis is required, which may be best addressed in a separate study. This comparison deserves a separate manuscript.

29. Section 5.6: largely repeating section 4.2.1, can be removed.

We removed this section.

We would like to thank the reviewer again for valuable comments, which helped us improve the quality of the manuscript.

Sincerely,

Yury Alkhimenkov, Lyudmila Khakimova and Yury Podladchikov

---

## Author Response (AR2)

Dear authors,

I have heard back from the two original referees, and while you have addressed many points they raised, both referees still have brought up important concerns that should be addressed before the manuscript can be accepted. The most important points are the convergence of the results, theoretical estimates for the resolution that would be required for convergence, and the effect of regularization viscosity on structural weakening, as well as an explanation for the change in initial conditions and the corresponding update of the results.

I understand that the stabilization method involves changing the viscosity at the same time as the mesh is refined, but that still allows for performing an additional resolution test in which the viscosity is kept constant to demonstrate that the solution does not substantially change with a finer mesh for that given viscosity. If converged results are not feasible for all cases, it would be useful for the readers to have a framework outlining what resolution would be required in comparison to the current resolution, an overview over the required computational resources for that resolution, and a justification for why the presented models capture the relevant physical mechanisms even if the results still change with increasing resolution.

Best regards,
Juliane Dannberg

Dear Dr. Dannberg, Dear reviewers,

Thank you for your detailed and constructive feedback. We have carefully considered both your comments and those from the reviewers, and have substantially revised the manuscript to address the key concerns.

Our original goal was to demonstrate spontaneous stress drops and strain localization in a minimalistic elasto-plastic model, supplemented by additional closely-related material on structural softening and the mechanics of stress release. However, we recognize the seriousness of the concerns raised, particularly regarding convergence and the possibility of numerical artifacts. These comments prompted a major revision and refocusing of the manuscript.

We admit that the reviewers suggested very high standards during the revision of this manuscript and two month (revision time) is not sufficient to address properly all the questions raised. For example, a single loading increment of a simulation with a resolution of N=8,000 grid cells takes around 1 day using a professional modern GPU. We need hundreds-thousands such loading increments to address some questions raised by reviewers. Therefore, we have removed some material from the original version of the manuscript and focus in the new version on only key novel aspects and converged setup.

In the updated version, we provide a clearly defined initial geometry for the 2D model and present a convergence study that spans a wide range of spatial resolutions—from N = 63^2 to N = 2047^2—while keeping the regularization viscosity constant across simulations. Based on this analysis, we identify $N = 1023^2$ as the minimum resolution at which both stress drop behavior and strain localization patterns converge for the considered simulation length. **All subsequent results in the manuscript, including the earthquake sequence and stress drop statistics, are based exclusively on these converged simulations.**

To ensure clarity and focus, we have removed several secondary and closely-related sections from the original version. To include these closely-related sections we need to run more simulations which will take several months or more, therefore, we will answer other questions in the following studies.

The revised manuscript now concentrates only on the core contributions:
**(1) convergence behavior**, and
**(2) the emergence of stress drops and earthquake-like sequences in a pressure-sensitive elasto-plastic medium.**

We note that we do not introduce a new model here but we adopt a regularization approach commonly used in geodynamic modeling (e.g., as proposed by Duretz et al. in Geophysical Research Letters) and apply it consistently across simulations.

We hope that these substantial revisions address the reviewers' concerns and meet the expectations for publication in Solid Earth. We appreciate your time and consideration, and we look forward to your response.

Best regards,

Yury Alkhimenkov, Lyudmila Khakimova and Yury Podladchikov

Response to Reviewer 1: Our comments are provided in blue. Text modifications are provided in green.

We would like to thank the reviewer again for valuable comments, which helped us improve the quality of the manuscript.

Major comments:

1. The authors made an effort to address my previous comments. The clarifications have exposed a major weakness of this work: the conclusions and discussions are based on numerical simulations that have not converged yet. In computational earthquake mechanics, conclusions are drawn from converged simulations. This has been the case at least since Jim Rice introduced in the 90's the distinction between continuum models and inherently discrete models. In the context of this work, converged simulations would be simulations in which the value of viscosity has been fixed and the spatial grid size and time step have been reduced (say, sequentially by a refinement factor of 2) until the difference between results of subsequently refined simulations become insignificant. In this manuscript, in many instances the viscosity is changed as the grid is refined and in other instances there is no evidence that the results have converged, which makes it impossible to draw conclusions. In very simple words: in the results presented, the reader cannot distinguish between meaningful results and numerical noise.

We thank the reviewer for this important observation. In the revised manuscript, we have addressed this concern by conducting a detailed and systematic convergence study. This study spans spatial resolutions from $63^2$ to $2047^2$, while keeping the regularization

viscosity fixed across all cases. We followed a standard refinement strategy by doubling the number of grid cells and observed that the differences between results diminish with increasing resolution.

Based on this analysis, we identified 1023^2 as the resolution at which both the stress drop amplitude and the strain localization patterns converge. All results and figures in the updated manuscript are based solely on these converged simulations. We believe that this now firmly addresses the issue of distinguishing between meaningful physical behavior and numerical artifacts.

2. In computational earthquake mechanics, in models based on fault friction, the resolution criterion is that the grid size should be much smaller than the size of the process zone (Day et al. 2005, https://doi.org/10.1029/2005JB003813). Analogously, for regularized plasticity, a natural criterion is that the grid size should be smaller than the thickness of shear bands. Some of the papers cited on regularized plasticity (e.g. by Duretz) might contain theoretical estimates of the thickness of shear bands as a function of the assumed viscosity and other model parameters, which can form a basis for a resolution criterion to guarantee convergence.

We appreciate the reviewer's suggestion regarding the use of theoretical estimates for shear band thickness as a resolution criterion. In our convergence study, we keep the regularization viscosity constant and observe that in the converged simulations (1023^2 and above), the thickness of shear bands spans more than 10 grid cells. This indicates that the regularization is functioning as intended, and that the localized deformation is well-resolved. Based on these converged results, we draw our conclusions regarding stress drops and earthquake sequence behavior.

3. Also, if theoretical studies are available about the effect of regularization viscosity on the existence of structural weakening, those would be important to shed light on your simulation results, from a more fundamental perspective.

We agree that it may be possible to derive theoretical relationships between regularization viscosity and shear band thickness, and that such analysis could provide deeper insights into structural weakening. However, this is beyond the scope of the present manuscript. Our focus here is on the dynamics of stress drops and earthquake sequences.

We adopt a regularization approach commonly used in geodynamic modeling (e.g., as proposed by Duretz et al. in Geophysical Research Letters) and apply it consistently across simulations. We do not introduce any novel regularization method or a completely new model. We base our conclusions on simulations that have demonstrably converged under this regularized framework. We believe this is sufficient for the current scope, which centers on emergent earthquake-like behavior in pressure-sensitive elasto-plastic media.

Minor comments:

1. In earthquake research the term "triggering" is associated with seismicity caused by a loading different than the slow tectonic loading, for instance by static or dynamic loading due to another earthquake, or by tides, hydrological loads, anthropogenic loads, etc. This topic is not treated in this paper, thus the word "triggering" should be replaced to avoid confusion. For example, in many instances, it can be replaced by "occurrence".

We have changed the title in the revised version.

Stress drop sequences in the simplest pressure-sensitive ideal elasto-plastic media: Implications for earthquake cycles

2. Lines 34-5 (of the pdf with tracked changes): As I noted in my previous review, Andrews (1976) introduced simulations with plasticity in the bulk. He was clearly very far ahead of his time. To my knowledge this was the first paper modeling earthquakes with off-fault plasticity. I think the paper deserves to be cited in that context too, not only as a paper introducing slip-weakening.

We appreciate the reviewer's suggestion and fully acknowledge that Andrews (1976) was an important and pioneering work. However, after carefully reviewing the paper, we note that while it does introduce the concept of slip-weakening and mentions plasticity in the bulk, it does not provide a rigorous investigation of off-fault plastic deformation in the modern sense. Specifically, the model lacks an analysis of strain localization, shear band formation, or systematic resolution testing to demonstrate the effects of bulk plasticity. In contrast, the current study explicitly focuses on the role of plastic strain localization, including convergence testing, visualization of shear bands, and quantification of stress drop behavior. For this reason, in the context of our discussion on elasto-plastic models and strain localization, we have cited Andrews (1976) as the first work to introduce slip-weakening—an idealization consistent with perfect plasticity—but do not include it as a detailed model of off-fault plasticity in the same sense as recent works on strain localization in geodynamic and earthquake modeling.

3. Line 38: Ma (2008) and Ma and Andrews (2010) should be cited instead in the next paragraph, as perhaps the earliest studies of dynamic rupture with plasticity in 3-D (as I noted in my previous review).

Thank you for the suggestion. As noted in our previous response regarding Andrews (1976), we acknowledge that Ma (2008) and Ma and Andrews (2010) are among the earlier studies to incorporate plasticity in 3D rupture simulations. However, similar to the 1976 study, these works do not explicitly focus on strain localization, convergence analysis, or the systematic resolution of shear bands. While they represent an important step in modeling plastic deformation in 3D, our study emphasizes detailed visualization and resolution testing of plastic strain localization, which is not the primary focus in the cited works. For this reason, we continue to reference these studies in the context of early plasticity-based rupture models but distinguish our contribution by focusing on resolved shear bands and converged earthquake sequences in elasto-plastic media.

4. The discrete version of the regularization is presented in equation 30, but (as noted in my previous review) the continuum visco-plasticity equations that defined the regularized rheology should also be presented. I believe these can be taken from the cited references (e.g. by Duretz) The best place to present them is in section 2.3.

Implemented.

For the case of regularized plasticity, the the yield function is defined as \citep{heeres2002comparison}:
\begin{equation}\label{A61H}

```
F(\tau, p) = \sqrt{J_2} - \sin(\varphi) p - \cos(\varphi) c - \eta^{\mathrm{vp}}
\dot{\lambda}.
\end{equation}
```

5. Section 3.4 is empty.

*Corrected.*

6. Line 186: define Go. Is it simply G?

*Corrected.*

7. Line 187: the background strain rate is defined only later, in equations 22 and 23. Reorganize the text in such a way that quantities are defined the first time they appear.

*Corrected.*

8. Line 194: I think you should remove "in the dimensionless framework" because equations 22 and 23 are not dimensionless.

*Corrected.*

9. Line 212: "integrated stress" appears before it has been defined. Reorganize the text to avoid that.

*Corrected.*

10. Lines 223-226, "The absence of stress drops in low-resolution simulations suggests that grid refinement is necessary ... In contrast, our sufficient resolution simulations with N = 1023^2 grid cells reveal several significant stress drops": I suspect this is due, more fundamentally, to the effect of viscosity on the existence of structural weakening. However, this cannot be disentangled in your manuscript because you keep changing viscosity when you refine the grid, and the reader cannot tell if your simulations have converged. I think the proper way to study this problem is to fix the viscosity and refine the grid until convergence, then change the viscosity and repeat, and finally only show the converged results for each value of viscosity.

*We performed exactly this experiment in the new version of the manuscript.*

11. Lines 229-230: Here you could introduce and explain the concept of structural weakening, with proper references to fundamental papers on the topic.

*We do not have this section in the new version of the manuscript. However, the references and the concept are still explained in the following section.*

12. Section 4.4.1: Has convergence been achieved? Show also a simulation with strain increment 2e-5. Convergence would manifest as a decrease in the difference between subsequent pairs of simulations with a same refinement ratio of 2.

We performed exactly this experiment in the new version of the manuscript. We report now only converged results.

13. Line 407, "to the inability of strain localization to continue growing in the prescribed direction": this needs more explanation.

In the updated manuscript we have removed this sentence. We note that the reviewer have raised many important questions but it requires substantial work to proof the answers. Therefore, we focus only on the results that are based on converged simulations.

Response to Reviewer 2: Our comments are provided in blue. Text modifications are provided in green.

We would like to thank the reviewer again for valuable comments, which helped us improve the quality of the manuscript.

Thank you for your responses. I believe the manuscript is nearly ready for publication, pending a few minor revisions. Below are my comments on the revised manuscript:

Line 206: I observed that the initial condition has been modified. Originally, the authors defined a Gaussian-style cohesion, which has now been replaced by a stepwise pressure condition. However, the results have not been updated to reflect this change. I suggest the authors provide an explanation of the modifications made in this revision.

Thank you for pointing this out, in the present manuscript we used a cohesion inclusion. This has now been corrected in the updated manuscript, and all figures and simulation results have been regenerated to reflect the correct setup

Section 4.1: It appears that the low-resolution simulation was also conducted with a lower temporal resolution. This is not explicitly stated. However, it is evident from Figure 6 that temporal resolution plays a significant role. I recommend clarifying this in the manuscript.

We do not have this section in the updated manuscript. We report now only converged results.

Line 215: Could the authors specify the value of x0?

Corrected.

($x_0 = L_x/4$)

Line 269: The purpose of this newly added paragraph is unclear. The authors should provide additional context for this section, or, if no further explanation can be provided, consider removing it.

We do not have this section in the updated manuscript.

Figure 7: I noticed that the simulation with N=1023 exhibits a sharper stress drop compared to the higher-resolution simulations. This discrepancy likely relates to the choice of the regularization parameter. It would be helpful to include a brief note on this in the main text or the figure caption.

Yes, you are absolutely right. However, in the new version of the manuscript we show only converged results and do no analyze how viscosity affect the shear bands (this is outside the scope of the study).

Figure 8: This section seems less conclusive compared to the previous two. I suggest the authors indicate which of the three simulations they consider the most effective, along with an explanation of why. Additionally, it is claimed in the introduction that using a zero regularization parameter leads to localization in a single pixel. However, this is not clearly reflected in the figures. I question the validity of this claim and recommend adding a note about this observation in the figure caption.

We do not have this figure in the updated manuscript. In the new version of the manuscript we show only converged results. We do not present "trends" in the new version of the manuscript.

Sincerely,
Yury Alkhimenkov, Lyudmila Khakimova and Yury Podladchikov

---

## Author Response (AR3)

Dear authors,

I would like to thank you for all the work you put into addressing the reviewers' comments. I'd like to give you a final opportunity to address the minor comments from this last round of reviews before the manuscript is published.

Best regards,

Juliane Dannberg

Dear Dr. Dannberg, Dear reviewers,

Thank you for your constructive feedback. We have carefully considered comments from the reviewer, and have revised the manuscript.

Best regards,

Yury Alkhimenkov, Lyudmila Khakimova and Yury Podladchikov

Response to Reviewer 1: Our comments are provided in blue. Text modifications are provided in green.

The authors have made a commendable effort to focus their manuscript on converged simulations.

Thank you for your positive feedback.

Figure 8 shows that the amplitudes of stress drops overall decrease with increasing loading strain. For instance, they are clearly very small after a strain of 1.2 %, and quite large before a strain of 0.4%. To show this more clearly, I recommend plotting the logarithm of stress drop amplitudes as a function of loading strain.

We agree with the reviewer and have add a figure of stress drop amplitudes as a function of loading strain.

Figure 9 shows a histogram of stress drop for the whole simulation. I recommend plotting also the histogram of the later portion of the simulation after a loading strain of 1.2 %. Is it almost symmetric? Is it approximately log-normal?

We added a figure (histogram of the later portion of the simulation after a loading strain of 1.2 %.). No, it is not log-normal.

Based on that, you can discuss whether the asymmetry of the current histogram (noted in Line 418) arises from the fact that stress drops become smaller with increasing loading strain.

In this study, we report only the results of a single converges simulation with the same boundary conditions. As can be seen the stress drops become smaller. But in reality boundary conditions are not the same but changing, therefore, for evolving boundary conditions the stress drops may not become smaller (but even become larger). To mentions this in the manuscript, we need to present another study, therefore, it is not discussed in the manuscript.

Minor comments (line numbers refer to the pdf with tracked changes):

1. Lines 6-7, "Finer temporal and spatial discretization leads to sharper stress drops and lower minimum stress values": Clarify that this statement holds for simulations that have not converged yet. Once convergence is reached, by definition, results no longer depend on discretization refinement.

**We added more details.**

for simulations that have not converged yet

2. Line 22: the word "triggering" should be replaced through the whole paper, for the reason explained in minor comment #1 of my previous review

**We agree with the reviewer. We replaced triggering.**

**nucleation**

3. Lines 36-37, "Recent studies ...": I would modify this sentence as "Numerical studies have suggested that plasticity plays a crucial role in earthquake rupture, particularly through off-fault plasticity mechanisms (e.g., Andrews (1976, 2005))". In fact, such studies are not recent, the first one is 50 years old.

**We agree with the reviewer. We modified the sentence.**

Numerical studies have suggested that plasticity plays a crucial role in earthquake rupture, particularly through off-fault plasticity mechanisms

4. Lines 39, "earliest studies": add "earliest 3-D studies" and move this sentence to Line 47, before "Another significant advancement ..."

**We agree with the reviewer. We modified and moved the sentence.**

\cite{ma2008physical, ma2010inelastic} conducted some of the earliest 3D studies on dynamic rupture with plasticity.

5. Line 62-63: You can add that an important goal is to achieve convergence of the numerical results and then focus on high-resolution (converged) simulations.

**We agree with the reviewer. We have added a sentence.**

An important goal is to ensure convergence of the numerical results, after which we focus on high-resolution, converged simulations.

6. Lines 72-73, "We propose a physics-based explanation for spontaneous stress drops": The stress drops in your simulations are produced by the known process of structural softening. Thus this item is not a "novelty of the present study". 7. Lines 74-75, "We do not prescribe any pre-existing faults; instead, new faults emerge spontaneously from the stress field": This is a feature of multiple previous modeling works too, thus not a "novelty of the present study".

We agree with the reviewer. Our main contribution that we explore the physics proposed in (a few) previous studies with very high spatial and temporal resolution, that was not possible before, that's why we can analyze histograms and have many faults. We modified the text and removed "novelties":

The distinct features of this study among other recent HPC simulations are:

...

- 2. We systematically investigate a previously proposed physics-based explanation for spontaneous stress drops using convergence-controlled, high-resolution GPU simulations, thereby extending the accessible resolution and fidelity of the theory. \\
- 3. We allow faults to emerge spontaneously from the evolving stress field, as it was done in a few previous studies. Our higher spatial/temporal resolution and GPU throughput produce a much larger population of emergent faults, enabling analyses that were previously intractable. \\
- 8. Line 83: replace "1..3" by "1,2,3"

We agree with the reviewer. We modified the sentence.

1,2,3

9. Equation 5: v k might need a superscript "eb"

We agree that some explanation is needed, therefore, we modified two equations in the text.

10. Equation 15: plus sign is repeated

We agree with the reviewer. We modified the formula.

11. Section 3.1: I understand that during simulations you switch between explicit solver (during fast deformation periods) and accelerated-pseudo-transient solver (during slow quasi-static deformation periods). Explain the criterion used for switching. Discuss whether the choices made in the switching criterion (for instance, threshold values) affect the statistics of small stress drop events.

APT solver is in fact a wave propagation solver. We simply add visualization of wave propagation for a displacement caused by a single stress drop. This visualization does not affect the quasi-static simulation.

The quasi-static equations are solved with the accelerated pseudo transient method (described below), while the dynamic wave-propagation is only visualized to illustrate the transient fields and does not affect the quasi-static evolution.

12. Line 227: define "N"

We agree with the reviewer. We modified the sentence.

(\$N\$ is the number of grid cells in x-dimension)

13. Legend of figure 2a: define "n\_x" (or did you mean N?)

We modified the figure. It should be N.

14. Lines 445-446, "the prevalence of small events and the presence of occasional larger ones are qualitatively consistent with the Gutenberg–Richter relationship": This statement conflates stress drop and earthquake magnitude. In natural earthquakes, stress drops are quite independent of magnitude. The Gutenberg-Richter relation pertains to magnitudes, not to stress drops.

We agree with the reviewer. We modified the sentence.

Gutenberg-Richter-like

15. Line 508, "low-resolution simulations, where the regularization length scale becomes comparable to or larger than the grid resolution": but this holds instead for high-resolution simulations, which have shear band thickness larger than the grid size. Clarify.

In large resolution simulation, with correct regularization, shear band thickens is a about several grid cells (e.g., 10-30). As we show in figure 2, resolution does play a key role.

16. There is dissonance between sections 6.2 and 6.2. The former emphasizes the importance of regularization to obtain well-resolved results, while the latter argues that similar results are obtained with and without regularization. Clarify.

Regularization is important for convergence of the results. But without regularization the results are similar --- same number of stress drops, approximately the same magnitudes.

17. Lines 524-525, "In 3D the stress and strain fields exhibit additional complexity, including the development of intersecting or branching shear bands": This features are also present in 2D, thus they are not an "additional complexity".

We agree with reviewer and removed this sentence.

18. Line 536, "quasi-periodic pattern": If you really mean quasi-periodicity, you should document it by computing the Coefficient of Variation (COV) of interevent times (the time intervals between stress drops). COV is defined as the standard deviation divided by the mean value. A small COV indicates quasi-periodic behavior.

We agree with reviewer and modified this sentence. The pattern of stress drops does not correspond to log-normal or normal distributions. This is a property of "hard" turbulence. But we do not mention it in the main text since it requires a specific study, focused on this property.

The pattern of stress drops does not correspond to log-normal or normal distributions.

19. Line 594: Are you suggesting that "subsequent stress drops" are not due to structural softening, but to a different mechanism? Clarify.

**We clarified this in the text.**

In this study, we investigated stress drops and earthquake-like behavior in idealized elastoplastic media using two-dimensional numerical simulations. The first stress drop occurs following the onset of strain localization, a process driven by structural softening \citep{vermeer1990orientation, le2013strain, sabet2019structural}. This structural softening mechanism, which received relatively little attention until recently \citep{sabet2019structural}, is explored here as a cause of spontaneous strain localization in an ideal plasticity model with a constant friction coefficient. Subsequent stress drops are associated with transitions between quasi-static loading intervals, during which the system moves from one equilibrium state to another due to the inability of strain localization to continue growing in the same direction. This behavior is consistent with fault offset theories developed by \cite{forsyth1992finite, buck1993effect} and validated by \cite{\larger{lavier1999self}. \cite{\forsyth1992finite} emphasized that Anderson's theory for faulting applies strictly to infinitesimal displacements. The initial orientation of a fault corresponds to the orientation that minimizes the regional stress required for slip initiation. However, \cite{forsyth1992finite} demonstrated that the additional horizontal stress necessary to maintain slip along the same fault increases linearly with accumulated displacement. Consequently, after only a few hundred meters of slip on a typical fault, it becomes mechanically more favorable to nucleate a new fault than to continue slip on the pre-existing one. Switching from sliding along an active fault to nucleation of a new fault is a fundamental cause of sudden stress drops and a potential mechanism for earthquake cvcles.

20. There are several typos. They can be fixed by running a spell/grammar checking tool.

We would like to thank the reviewer again for valuable comments, which helped us improve the quality of the manuscript.

Sincerely,

Yury Alkhimenkov, Lyudmila Khakimova and Yury Podladchikov